# A multisite validation of brain white matter pathways of resilience to chronic back pain

**Mina Mišić[1]\*[†], Noah Lee[2][†], Francesca Zidda[1], Kyungjin Sohn[3], Katrin Usai[1], Martin Löffler[1,4], Md Nasir Uddin[5], Arsalan Farooqi[2], Giovanni Schifitto[5], Zhengwu Zhang[3], Frauke Nees[1,6][‡], Paul Geha[2][‡], Herta Flor[1][‡]**

[1]Institute of Cognitive and Clinical Neuroscience, Central Institute of Mental Health, Medical Faculty Mannheim, Heidelberg University, Mannheim, Germany; [2]Department of Psychiatry, University of Rochester Medical Center, Rochester, United States; [3]Department of Statistics and Operations Research, University of North Carolina, Chapel Hill, Rochester, United States; [4]Department of Experimental Psychology, Heinrich Heine University Düsseldorf, Düsseldorf, Germany; [5]Department of Neurology, University of Rochester Medical Center, Rochester, United States; [6]Institute of Medical Psychology and Medical Sociology, University Medical Center Schleswig Holstein, Kiel University, Kiel, Germany

\*For correspondence: mina.kandic@zi-mannheim.de

[†]These authors contributed equally to this work

[‡]These authors also contributed equally to this work

## eLife Assessment

This **valuable** study provides **convincing** evidence that white matter diffusion imaging of the right superior longitudinal fasciculus might help to develop a predictive biomarker of chronic back pain chronicity. The results are based on a discovery-replication approach with different cohorts, but the sample size is limited. The findings will interest researchers interested in the brain mechanisms of chronic pain and in developing brain-based biomarkers of chronic pain.

**Abstract** Chronic back pain (CBP) is a global health concern with significant societal and economic burden. While various predictors of back pain chronicity have been proposed, including demographic and psychosocial factors, neuroimaging studies have pointed to brain characteristics as predictors of CBP. However, large-scale, multisite validation of these predictors is currently lacking. In two independent longitudinal studies, we examined white matter diffusion imaging data and pain characteristics in patients with subacute back pain (SBP) over 6- and 12-month periods. Diffusion data from individuals with CBP and healthy controls (HC) were analyzed for comparison. Whole-brain tract-based spatial statistics analyses revealed that a cluster in the right superior longitudinal fasciculus (SLF) tract had larger fractional anisotropy (FA) values in patients who recovered (SBPr) compared to those with persistent pain (SBPp), and predicted changes in pain severity. The SLF FA values accurately classified patients at baseline and follow-up in a third publicly available dataset (Area under the Receiver Operating Curve ~0.70). Notably, patients who recovered had FA values larger than those of HC suggesting a potential role of SLF integrity in resilience to CBP. Structural connectivity-based models also classified SBPp and SBPr patients from the three data sets (validation accuracy 67%). Our results validate the right SLF *as a robust predictor of CBP development, with potential for clinical translation.* Cognitive and behavioral processes dependent on the right SLF, such as proprioception and visuospatial attention, should be analyzed in subacute stages as they could prove important for back pain chronicity.

## Introduction

Chronic back pain (CBP) is the leading cause of disability worldwide *Hoy et al., 2012*; the annual contribution of patients with back pain to all-cause medical costs is $365 billion in the United States alone (*Lo et al., 2021*). In 2020, there were over 500 million reported cases of prevalent low back pain globally and central Europe had the highest age-standardized rate of prevalence with 12,800 cases per 100,000 individuals (*Ferreira et al., 2023*). In addition, CBP is associated with significant co-morbidities such as substance misuse, (*Deyo et al., 2015*) depression, (*Gore et al., 1976*; *Becker, 1976*; *Wasan et al., 2015*) anxiety, (*Gore et al., 1976*; *Wasan et al., 2015*) and obesity (*Shiri et al., 2010*; *Shiri et al., 2014*). The large majority of CBP cases suffer from primary pain (*Finucane et al., 2020*). It is estimated that 25–50% of patients with subacute back pain (SBP, duration 6–12 weeks) go on to develop CBP (*da C Costa et al., 2009*; *Stevans et al., 2021*; *Nieminen et al., 2021*; *Axén and Leboeuf-Yde, 2013*; *Price et al., 2018*). Once back pain is chronic, it is very difficult to treat (*Machado et al., 2010*; *Henschke et al., 2010*; *Costa et al., 2009*; *Wand and O'Connell, 2008*; *da C Menezes Costa et al., 2012*). Therefore, prevention of the transition to the chronic phase (*Traeger et al., 2016*) is key to reducing the prevalence, and as a result, the burden of CBP.

Early identification of the likelihood of pain chronicity is a pre-requisite step for allowing the efficient implementation of preventive treatment (*Berwick and Hackbarth, 2012*) because diagnostic and treatment procedures offered to all (sub)acute back pain patients expose resilient patients to unnecessary and risky procedures. In addition, treatment offered to all patients is burdensome to the healthcare system (*Litkowski et al., 2016*; *R. Chou, 2011*). However, validated quantitative prognostic biomarkers that can predict disease trajectory and guide treatment in back pain patients have yet to be identified.

Some studies have addressed this question with prognostic models incorporating demographic, pain-related, and psychosocial predictors (*Traeger et al., 2016*; *Hill et al., 2008*; *Hockings et al., 2008*; *Chou and Shekelle, 2010*). While these models are of great value showing that few of these variables (e.g. work factors) might have significant prognostic power on the long-term outcome of back pain, their prognostic accuracy is limited (*Silva et al., 2022*) with parameters often explaining no more than 30% of the variance (*Kent and Keating, 2008*; *Hruschak and Cochran, 2018*; *Hartvigsen et al., 2018*). A recent notable study in this regard developed a model based on easy-to-use brief questionnaires to predict the development and spread of chronic pain in a variety of pain conditions capitalizing on a large dataset obtained from the UK-BioBank (*Tanguay-Sabourin et al., 2023*). This work demonstrated that only a few features related to the assessment of sleep, neuroticism, mood, stress, and body mass index were enough to predict persistence and spread of pain with an area under the curve of 0.53–0.73. Yet, this study is unique in showing such a predictive value of questionnaire-based tools. Neurobiological measures could therefore complement existing prognostic models based on psychosocial variables to improve overall accuracy and discriminative power. More importantly, neurobiological factors such as brain parameters can provide a mechanistic understanding of chronicity and its central processing.

Neuroimaging research on chronic pain has uncovered a shift in brain responses to pain when acute and chronic pain are compared. The thalamus, primary somatosensory, motor areas, insula, and mid-cingulate cortex most often respond to acute pain and can predict the perception of acute pain (*Wager et al., 2013*; *Lee et al., 2021*; *Becker et al., 2018*; *Spisak et al., 2020*). Conversely, limbic brain areas are more frequently engaged when patients report the intensity of their clinical pain (*Baliki et al., 2015*; *Elman and Borsook, 2016*). Consistent findings have demonstrated that increased prefrontal-limbic functional connectivity during episodes of heightened subacute ongoing back pain or during a reward learning task is a significant predictor of CBP (*Baliki et al., 2012*; *Löffler et al., 2022*). Furthermore, low somatosensory cortex excitability in the acute stage of low back pain was identified as a predictor of CBP chronicity (*Jenkins et al., 2019*). One study so far has investigated structural changes in back pain using a longitudinal design and found that changes across several white matter tracts including the temporal part of the left superior longitudinal fasciculus, external capsule, parts of the corpus callosum, and parts of the internal capsule were predictive of back pain development at 1-year follow-up (*Mansour et al., 2013*). Despite the identification of these promising biomarkers of pain chronicity, their validation across multiple independent cohorts remains rare (*Davis et al., 2020*). Here, we investigated brain white matter predictors of back pain chronicity across three independent samples originating from 3 different sites and expected white matter tracts previously

found predictive of CBP *Mansour et al., 2013* to show higher fractional anisotropy values (FA; greater structural integrity) in patients who recovered compared to those whose pain persisted. We additionally hypothesized that patients who recovered would also exhibit greater structural integrity within the prefrontal-limbic tract related to learning and memory, uncinate fasciculus (*Von Der Heide et al., 2013*), in line with findings that learning-related processes could be predictive of CBP (*Löffler et al., 2022*). We expected that lower FA at baseline would predict persistent state and greater pain severity at follow-up.

## Results

### Demographic and clinical characteristics

At baseline, the data collected in New Haven included 16 SBP patients who recovered (SBPr) at approximately 1-year follow-up as their low-back pain intensity dropped by more than 30% relative to baseline and 12 SBP patients whose pain persisted at follow-up. The SBPp patients were older (38.0±3.6 years, average ± SEM) and had pain for a slightly longer duration (10.8±0.9 weeks) than the SBPr patients (age = 30.8 ± 2.2 years; pain duration = 8.6 ± 0.9 weeks) but these differences did not reach statistical significance (t-score (degrees of freedom) ($t$(df)) = 1.8 (26), p = 0.08 for age and $t$(df) = 1.8 (26), p = 0.08 for duration comparisons, respectively, unpaired T-test). The groups did not significantly differ in the distribution of males and females ($\chi2$ = 0.32, df = 1, p = 0.57) or body mass index (BMI) ($t$(df) = 0.35 (26), p = 0.73). They did not significantly differ in average reported pain intensity ($t$(df) = - 0.45 (*Hockings et al., 2008*), p = 0.66). The SBPr patients reported significantly (p = 0.02) larger depression scores on Beck's Depression Inventory (BDI) (7.3±1.2) than the SBPp patients (3.1±1.1, $t$(df) = - 2.65 (*Hockings et al., 2008*), p = 0.02) but the average score indicated that the SBPr patients did not have any clinically significant symptoms (i.e. average BDI <10).

### Whole-brain tract-based spatial statistics

We calculated voxel-wise FA of each participant and performed a whole brain comparison over the white matter skeleton using permutation testing between SBPr (n=16) and SBPp (n=12) patients at baseline (unpaired t-test, p<0.05, cluster-based thresholding) corrected for age, gender, and head displacement estimated by eddy current correction (*Figure 1—figure supplement 1*). SBPr showed larger FA within the cluster of fibers part of the right superior longitudinal fasciculus (SLF; MNI-coordinates of peak voxel: x=35; y = - 13; z=26 mm; t(max)=4.61; *Figure 1*). The possibility that the observed difference between SBPr and SBPp patients was due to a difference in the amount of displacement applied during registration was ruled out (*Figure 1—figure supplement 1*). We plotted the FA values of the same SLF region from healthy controls (HC) and CBP patients; *Figure 1* shows FA values and their distributions for each group within the SLF cluster. SBPr patients had the largest FA values in the right SLF cluster, even larger than in HC, although this difference did not reach statistical significance (p=0.11). We also examined mean diffusivity (MD), axial diffusivity (AD), and radial diffusivity (RD) extracted from the right SLF shown in *Figure 1* to further understand which diffusion component is different between the groups. The right SLF MD is significantly increased (p<0.05) in the SBPr compared to SBPp patients (*Figure 1—figure supplement 2*), while the right SLF RD is significantly decreased (p<0.05) in the SBPr compared to SBPp patients in the New Haven data (*Figure 1—figure supplement 3*). Axial diffusivity extracted from the RSLF mask did not show significant difference between SBPr and SBPp (p=0.28; *Figure 1—figure supplement 4*).

To test whether baseline FA values predict the change in pain severity from baseline to follow-up in the New Haven data set, we did multiple regression analysis with FA values as predictor and the percentage change in pain severity as outcome. In this model, FA values were predictive of pain severity at the 1-year follow-up (adjusted $R^2$ = 0.202, p = 0.009). To confirm that this result was not driven by age, gender, or head motion, we entered these parameters in a new model adjusting the prediction for these covariates. FA values were still predictive of the change in pain intensity, with added variables improving the model fit (new model: adjusted $R^2$ = 0.259, p = 0.037; difference between models: $F$(2,67) = 1.50, p=0.237). *Figure 2* depicts the correlation between FA values in the right SLF and pain severity with higher FA values (greater structural integrity) associated with greater reduction in pain (percentage change).

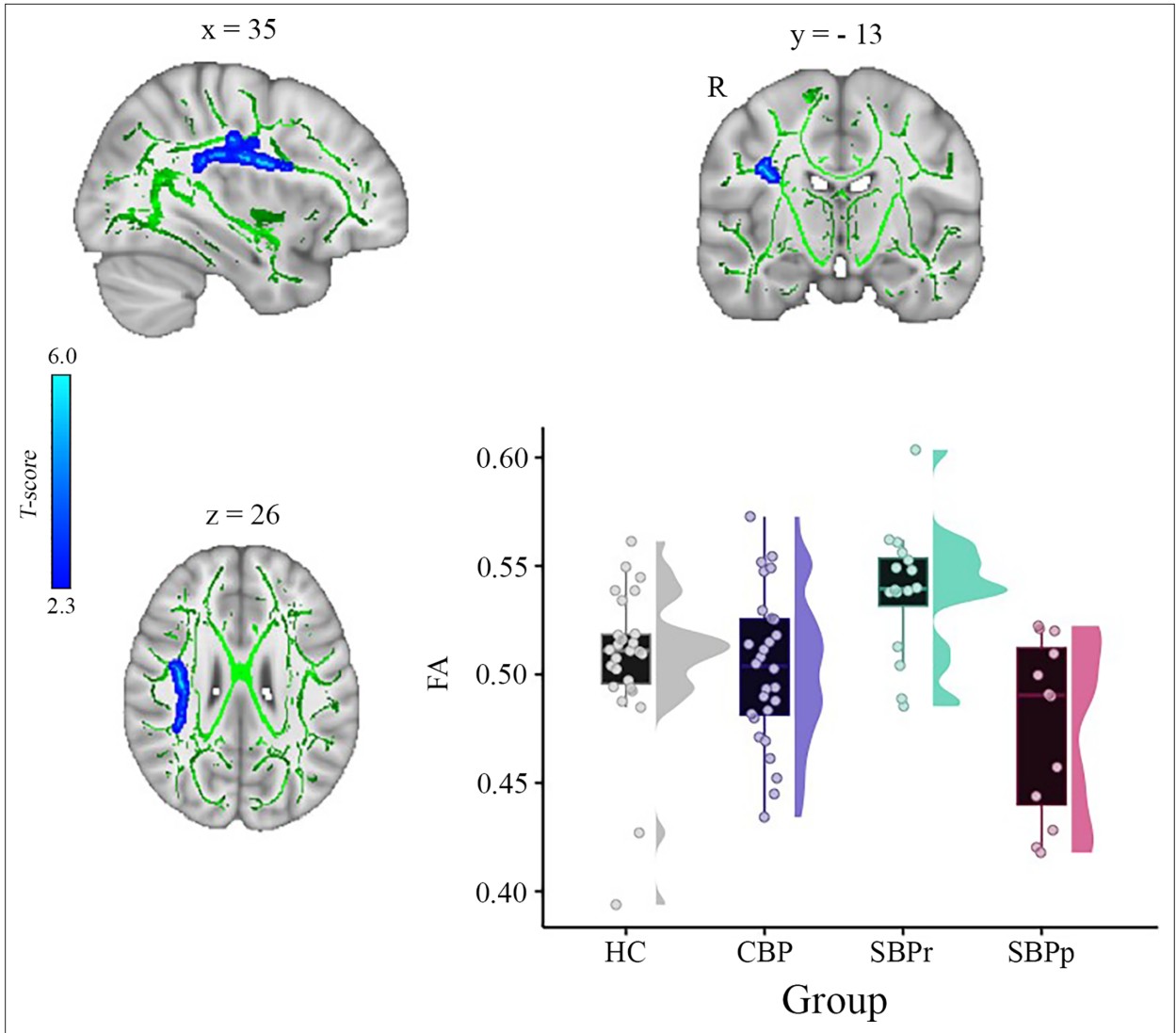

**Figure 1.** A whole brain comparison over the white matter skeleton between SBPr and SBPp patients at baseline and distribution of FA values for each group (*New Haven data set*). Results of unpaired t-test (p<0.05, 10,000 permutations) showing significantly increased FA (fractional anisotropy) in SBPr (recovered) compared to SBPp (persistent) patients within the right superior longitudinal fasciculus (SLF) in the New Haven data set. Rain clouds include boxplots and the FA data distribution for each group depicted on the right side of each boxplot. Jittered circles represent single data points, the middle line represents the median, the hinges of the boxplot the first and third quartiles, and the upper and lower whiskers 1.5*IQR (the interquartile range).

The online version of this article includes the following figure supplement(s) for figure 1:

**Figure supplement 1.** Motion parameters in the Discovery data set.

**Figure supplement 2.** Distribution plots of mean diffusivity (MD) extracted from the RSLF mask shown in *Figure 1*.

**Figure supplement 3.** Distribution plots of radial diffusivity (RD) extracted from the RSLF mask shown in *Figure 1*.

**Figure supplement 4.** Distribution plots of axial diffusivity (AD) extracted from the RSLF mask shown in *Figure 1*.

**Figure supplement 5.** TBSS analysis and RSLF FA validation after applying neuroCombat after pooling subjects from the three sites into one analysis.

**Figure supplement 6.** Plot of FA values before (black) and after (red) applying harmonization using neuroCombat calculated as mean skeletal FA.

## Validation of the results obtained from the New Haven data
### Mannheim data

In another independent study, a whole-brain comparison of FA over the white matter skeleton using permutation testing unpaired t-test, $P<0.05$, threshold-free cluster enhancement (TFCE) corrected for age, gender, and two motion parameters (translation and rotation) revealed two clusters, one in the right superior longitudinal fasciculus (SLF) tract (cluster size = 409 voxels, MNI-coordinates of peak

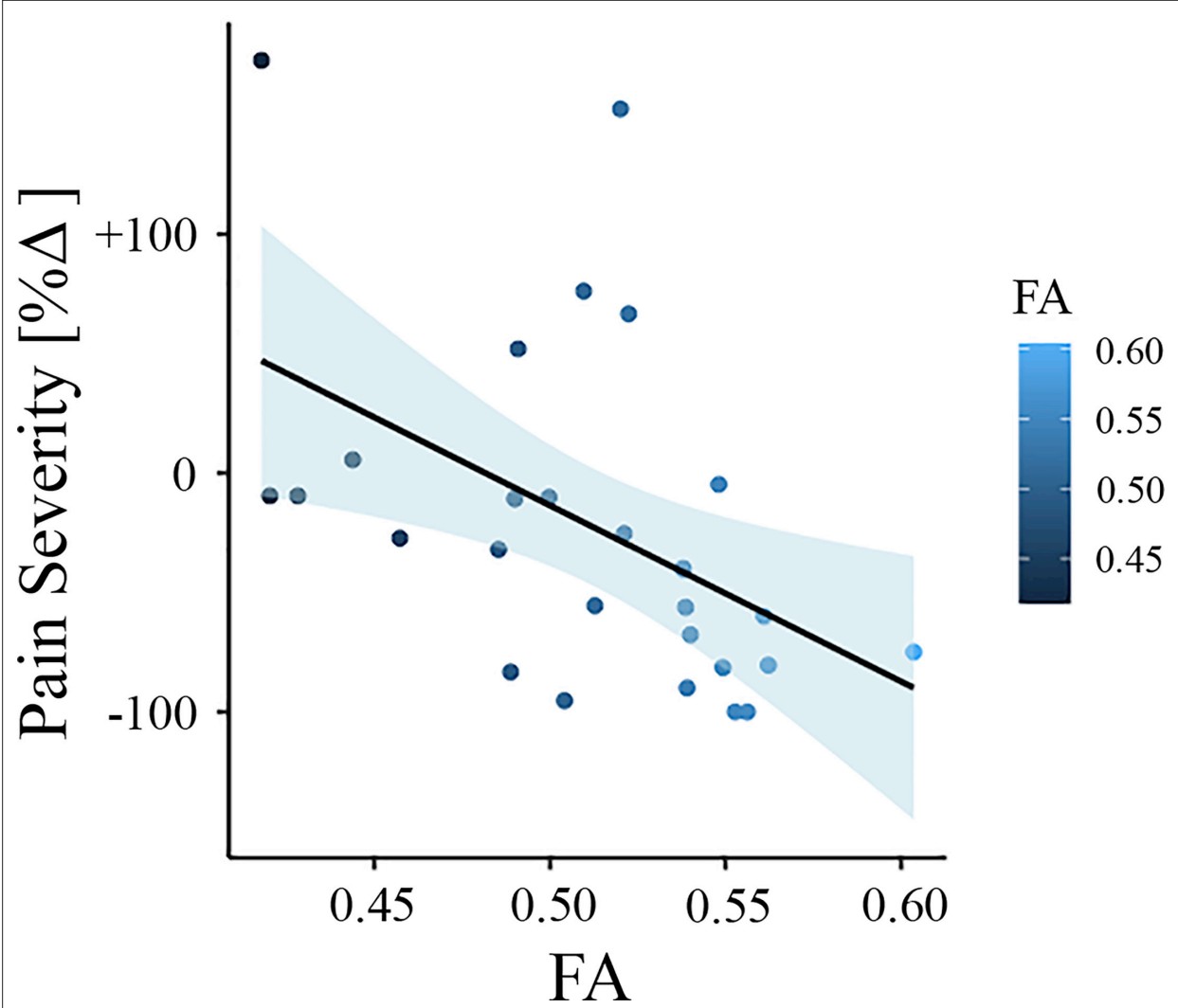

**Figure 2.** Association between white matter FA values and pain severity (*New Haven data set*). Higher fractional anisotropy (FA) values in the right superior longitudinal fasciculus (SLF) are associated with greater pain reduction (from baseline to follow-up) in the New Haven data set.

voxel: x = 26, y = –33, z = 45, *p*(TFCE) = 0.041, t(max) = 3.57) and one in the right corticospinal tract/superior corona radiata (cluster size = 381 voxels, MNI-coordinates of peak voxel: x = 29, y = –16, z = 21, *p*(TFCE) = 0.041, t(max) = 3.63) that were significantly greater in SBPr (N = 28) compared to SBPp (N=18) (*Figure 3*). In addition, two smaller clusters were also identified in the same tract of the right SLF (cluster size = 39 voxels, MNI-coordinates of peak voxel: x = 36, y = –13, z = 34, *p*(TFCE) = 0.048, t(max) = 3.19) and right corticospinal tract (cluster size = 13 voxels, MNI-coordinates of peak voxel: x = 21, y = –27, z = 42, *p*(TFCE) = 0.049, t(max) = 2.41) as significantly different between SBPr and SBPp patients. In the next step, we extracted FA values from the significant clusters of the Mannheim data and compared them across all groups at that site. As in the New Haven set, recovered patients had the largest FA values across groups, even greater than HC although this difference did not reach significance level (p = 0.12; *Figure 3*).

To test whether FA baseline values from the significantly different clusters could predict the change in pain severity from baseline to follow-up, we did multiple regression analysis with FA values as predictor and the change in pain severity percentage as outcome. In this model, FA values were predictive of pain severity at the 6 month follow-up (adjusted $R^2$ = 0.120, p = 0.011). To confirm that this result was not driven by age, gender, or head motion, we entered these parameters in a new model adjusting the prediction for covariates. FA values were still predictive of chronicity, with added variables improving the model fit (new model: adjusted $R^2$ = 0.236, p = 0.007; difference between

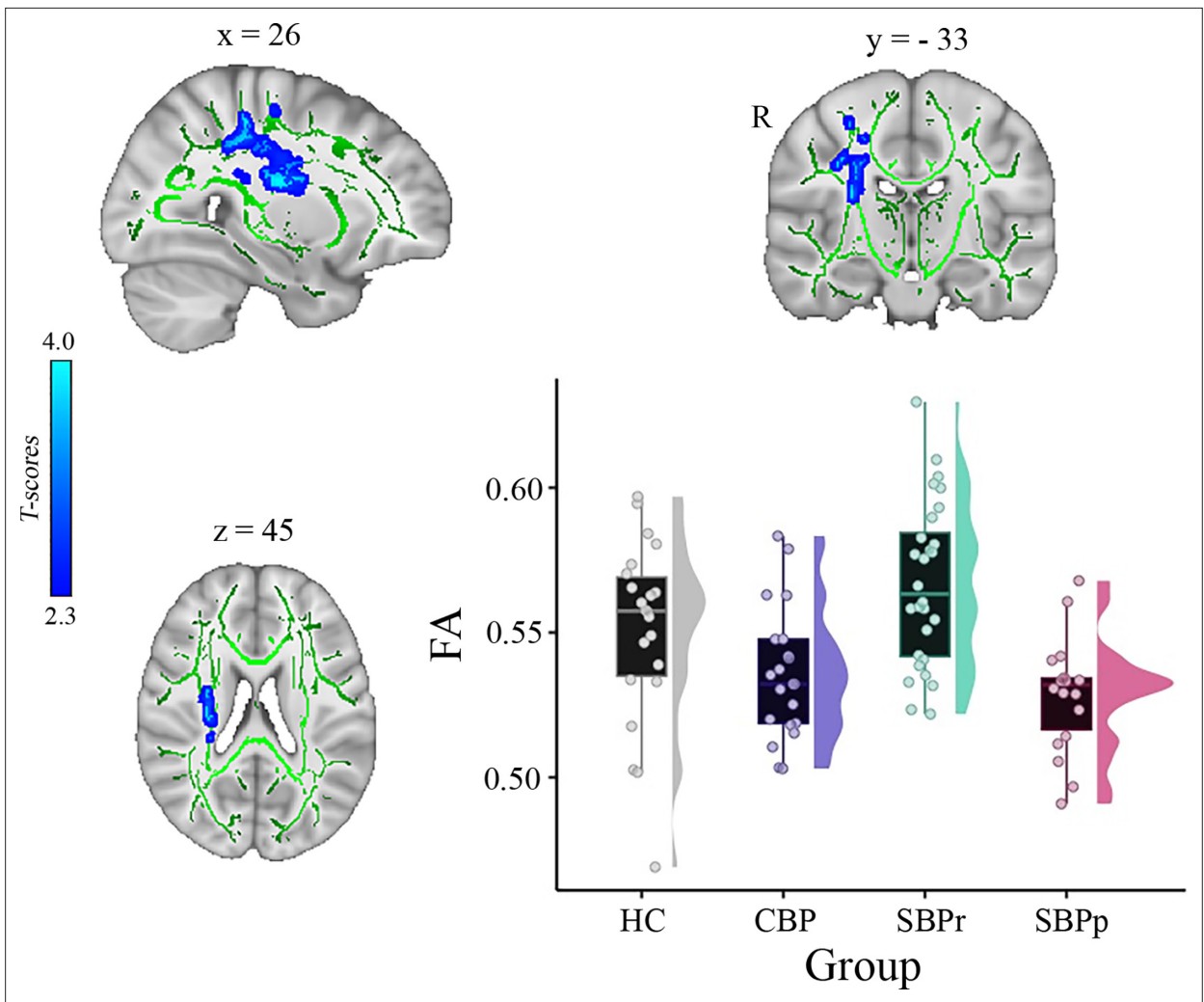

**Figure 3.** A whole brain comparison over the white matter skeleton between SBPr and SBPp patients at baseline and distribution of FA values for each group (*Mannheim data set*). Results of unpaired t-test (p<0.05, 10,000 permutations) showing significantly increased FA (fractional anisotropy) in SBPr (recovered) compared to SBPp (persistent) patients at six-months follow-up within the right superior longitudinal fasciculus (SLF) in the Mannheim data set. Rain clouds include boxplots and the FA data distribution for each group depicted on the right side of each boxplot. Jittered circles represent single data points, the middle line represents the median, the hinges of the boxplot the first and third quartiles, and the upper and lower whiskers 1.5*IQR (the interquartile range).

The online version of this article includes the following figure supplement(s) for figure 3:

**Figure supplement 1.** Distribution of the values of the area under the curve (AUC) when shuffling the label of the groups (recovered, persistent) using the FA values from the right SLF before neuroCombat harmonization for the Mannheim data.

**Figure supplement 2.** Distribution of the values of the AUC when shuffling the label of the groups (recovered, persistent) using the FA values from the right SLF after neuroCombat harmonization for the Mannheim data.

models: $F(2,67) = 3.67$, p = 0.046). *Figure 4* depicts the correlation between FA values and pain severity with higher FA values (greater structural integrity) associated with greater reduction in pain (percentage change).

We tested the accuracy of local diffusion properties of the right SLF extracted from the mask of voxels passing threshold in the New Haven data (*Figure 1*) in classifying the Mannheim patients into persistent and recovered. We used a simple cut-off (*Green and Swets, 1966*) for the evaluation of the area under the receiver operating characteristic (ROC) curve (AUC). FA values corrected for age, gender, and head displacement, accurately classified SBPr (N = 28) and SBPp (N = 18) patients from the Mannheim data set with an AUC = 0.66 (p = 0.031, tested against 10,000 random permutations, see *Figure 3—figure supplement 1*), validating the predictive value of the right SLF cluster (*Figure 5*).

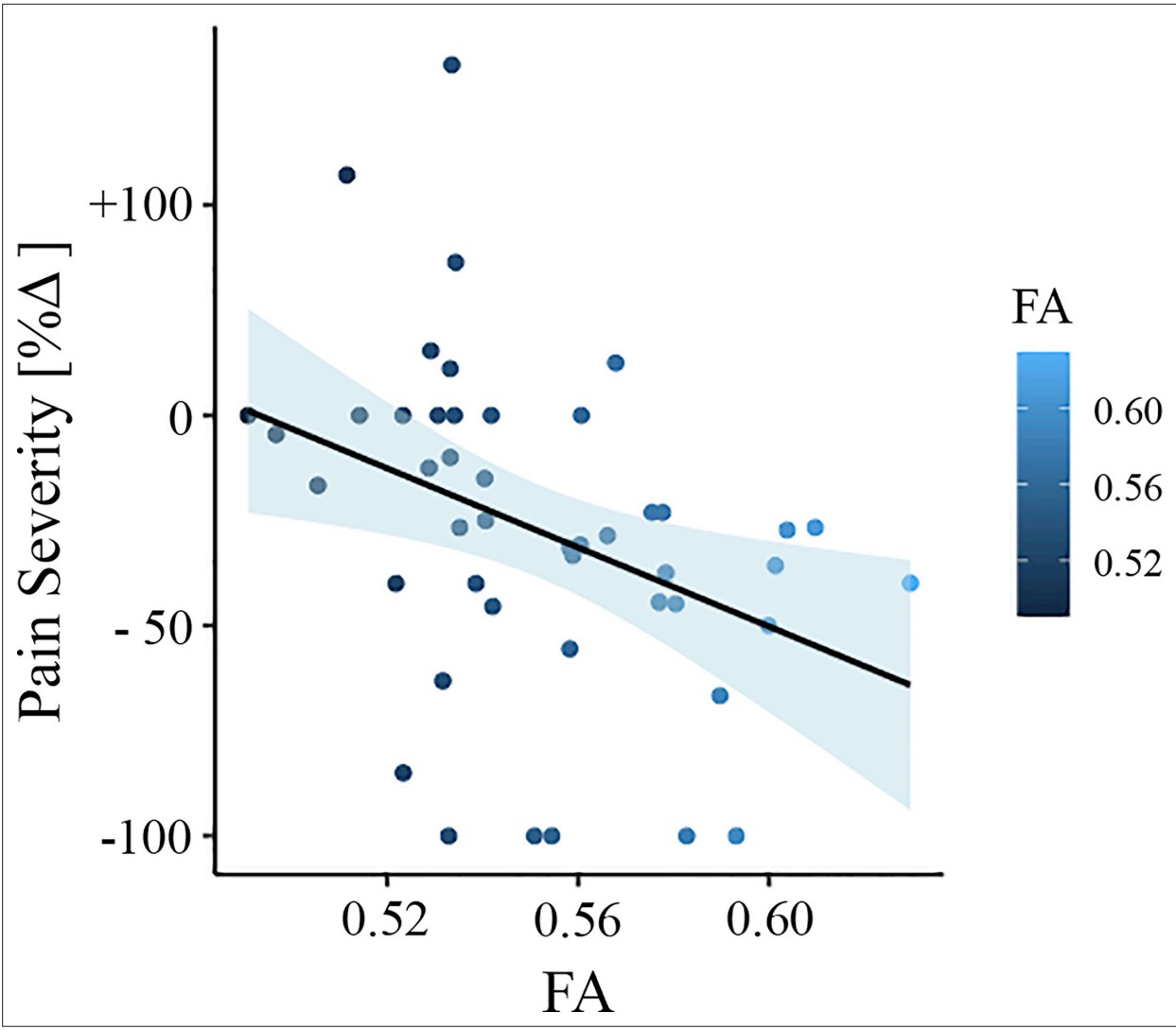

**Figure 4.** Association between white matter FA values and pain severity (*Mannheim data set*). Higher fractional anisotropy (FA) values in the right superior longitudinal fasciculus (SLF) are associated with greater pain reduction (from baseline to follow-up) in the Mannheim data set.

*Figure 5—figure supplement 1* shows the results in the Mannheim data set if a 30% reduction is used as a recovery criterion in this dataset (AUC = 0.53).

## Chicago data

To further validate the right SLF predictive power, we used the mask shown in *Figure 1* to extract FA values from SBPr and SBPp patients available in the Chicago data set. FA values in the validation data sets were corrected for age, gender, and head displacement. The criterion for recovery was set identically to the New Haven study, requiring ≥ 30% reduction of reported low-back pain intensity at one-year follow-up. FA values were calculated for two time points; one obtained at baseline when pain was still subacute (6–12 weeks), and one obtained at a one-year follow-up when pain either remitted or persisted (*Vachon-Presseau et al., 2016*). FA values of the right SLF (*Figure 1*) accurately classified SBPr (N = 23) and SBPp (N = 35) patients from Chicago with an AUC = 0.70 (p = 0.0043, *Figure 6—figure supplement 1*) at baseline (*Figure 6A*), and SBPr (N = 28) and SBPp (N = 34) patients with an AUC = 0.66 (p = 0.014, see *Figure 6—figure supplement 2*) at follow-up (*Figure 6B*), validating the predictive cluster from the right SLF at yet another site. The correlation between FA values in the right SLF and pain severity in the Chicago data set showed marginal significance (p = 0.055) at visit 1

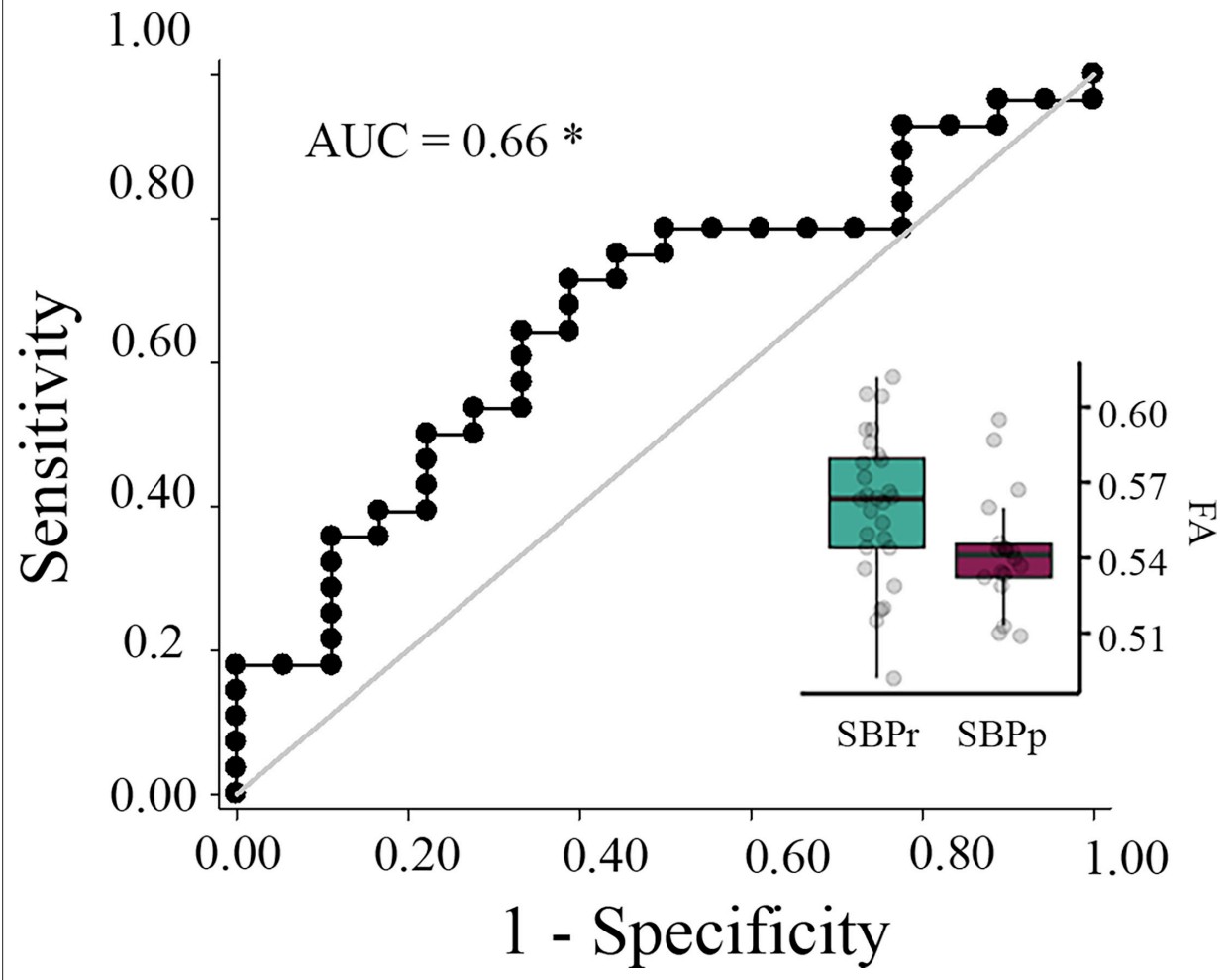

**Figure 5.** Validation of the accuracy of FA in the right SLF in classifying Mannheim patients. The right superior longitudinal fasciculus (SLF) cluster from the discovery set accurately classifies patients who recovered (SBPr) and those whose pain persisted (SBPp) in the Mannheim data set at a 6-month follow-up. Classification accuracy is based on the ROC curve. Circles on the boxplots represent single data points, the middle line represents the median, the hinges of the boxplot the first and third quartiles, and the upper and lower whiskers 1.5*IQR (the interquartile range). AUC: area under the curve; * p<0.05.

The online version of this article includes the following figure supplement(s) for figure 5:

**Figure supplement 1.** The receiver operating characteristic curve shows the area under the curve (AUC = 0.53) when using the mask from the New Haven data to classify the SBPr and SBPp patients from Mannheim using the 30% recovery criterion.

(*Figure 6—figure supplement 3*, panel A) and higher FA values were significantly associated with a greater reduction in pain at visit 2 (p = 0.035; *Figure 6—figure supplement 3*, panel B).

## Validation after harmonization

Because the DTI data sets originated from three sites with different MR acquisition parameters, we repeated our TBSS and validation analyses after correcting for variability arising from site differences using DTI data harmonization as implemented in neuroCombat (*Fortin et al., 2017*). The method of harmonization is described in detail in the Materials and methods section. The whole brain unpaired t-test depicted in *Figure 1* was repeated after neuroCombat and yielded very similar results (*Figure 1—figure supplement 5*, panel A) showing significantly increased FA in the SBPr compared to SBPp patients in the right superior longitudinal fasciculus (MNI-coordinates of peak voxel: x = 40; y = - 42; z = 18 mm; t(max) = 2.52; p<0.05, corrected against 10,000 permutations). We again tested the accuracy of local diffusion properties (FA) of the right SLF extracted from the mask of voxels passing threshold in the New Haven data (*Figure 1—figure supplement 5*, panel A) in classifying the

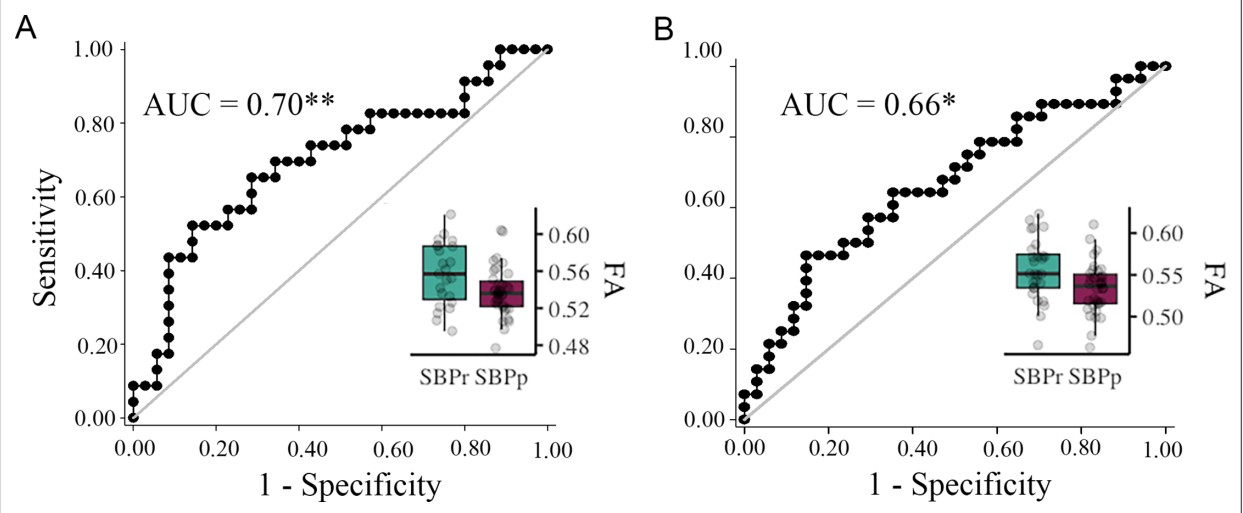

**Figure 6.** Validation of the accuracy of FA in the right SLF in classifying Chicago patients. The right SLF (superior longitudinal fasciculus) cluster from the discovery set accurately classifies patients who recovered (SBPr) and those whose pain persisted (SBPp) in the Chicago (OpenPain) data set at visit 1 (baseline) (**A**) and visit 2 (one-year follow-up) (**B**). Classification accuracy is based on the ROC curve. Circles on the boxplots represent single data points, the middle line represents the median, the hinges of the boxplot the first and third quartiles, and the upper and lower whiskers 1.5*IQR (the interquartile range). AUC: area under the curve; * p<0.05.

The online version of this article includes the following figure supplement(s) for figure 6:

**Figure supplement 1.** Distribution of the values of the area under the curve (AUC) when shuffling the label of the groups (recovered, persistent) using the FA values from the right SLF before neuroCombat harmonization for the Chicago data set at visit 1.

**Figure supplement 2.** Distribution of the values of the area under the curve (AUC) when shuffling the label of the groups (recovered, persistent) using the FA values from the right SLF before neuroCombat harmonization for the Chicago data set at visit 2.

**Figure supplement 3.** Correlation between FA values and pain severity with higher FA values (greater structural integrity) associated with the greater reduction in pain (percentage change) at baseline (n=58) (**A**) and follow-up (n=60) (**B**) in the Chicago data set.

**Figure supplement 4.** Distribution of the values of the area under the curve (AUC) when shuffling the label of the groups (recovered, persistent) using the FA values from the right SLF after neuroCombat harmonization for the Chicago data set at visit 1.

**Figure supplement 5.** Distribution of the values of the area under the curve (AUC) when shuffling the label of the groups (recovered, persistent) using the FA values from the right SLF after neuroCombat harmonization for the Chicago data set at visit 2.

Mannheim and the Chicago patients, respectively, into persistent and recovered. FA values corrected for age, gender, and head displacement accurately classified SBPr and SBPp patients from the Mannheim data set with an AUC = 0.67 p = 0.023, tested against 10,000 random permutations, *Figure 1— figure supplement 5*, panel B and *Figure 3—figure supplement 2*, and patients from the Chicago data set with an AUC = 0.69 (p = 0.0068), *Figure 1—figure supplement 5*; panel C and *Figure 6— figure supplement 4* at baseline, and an AUC = 0.67 (p = 0.0098) (*Figure 1—figure supplement 5*; panel D and *Figure 6—figure supplement 5*) patients at follow-up, confirming the predictive cluster from the right SLF across sites. The application of neuroCombat significantly changes the FA values as shown in *Figure 1—figure supplement 6* but does not change the results between groups.

## Structural connectivity-based classification of SBPp and SBPr patients

We studied the structural connectivity of brain areas known to be connected by different parts of the right SLF by illustrating in three dimensions the white matter fiber-tracts traveling between them (*Figure 7A*). We also extracted the FA values along those tracts (*Figure 7B*). Most of the visualized white matter bundles showed thinning (i.e., decreased density) in the SBPp patients when visually inspected relative to the SBPr patients (*Figure 7A*; *Figure 7—figure supplement 1*). Additionally, there was a drop in FA along the fiber tracts in the SBPp compared to the SBPr patients with a trend towards a lower number of tracts in the former group (*Figure 7B*).

Furthermore, using fiber count and total connected surface area connecting each pair of regions of the structural white matter connectome as the input features, we pooled the data from the three sites and machine learning cross-validation to classify sub-acute back pain patients as SBPr or SBPp. The

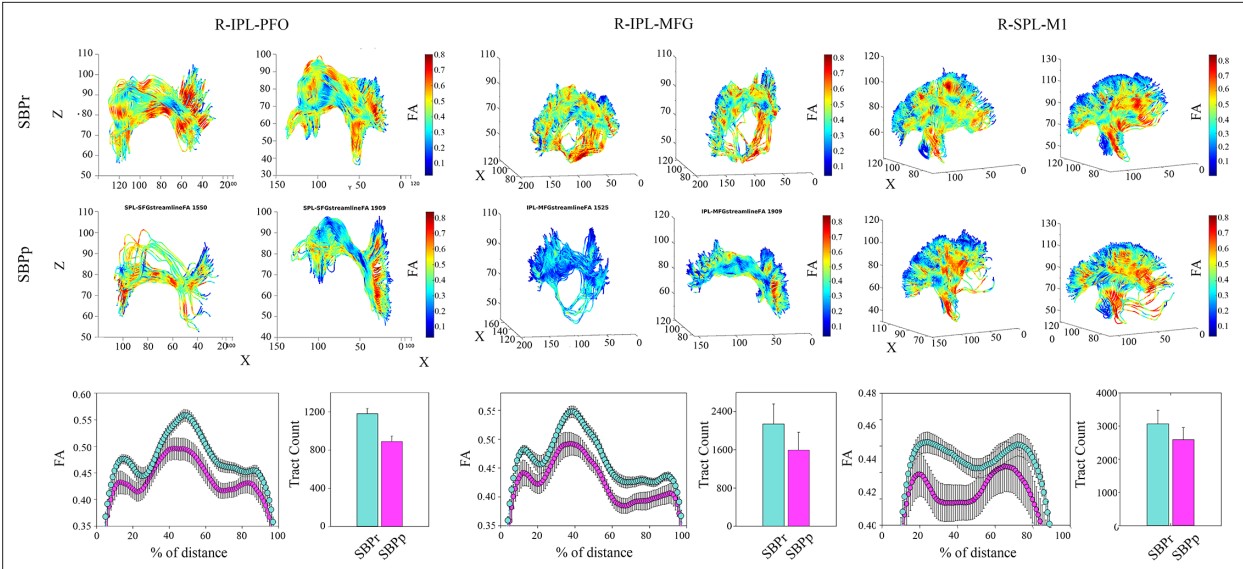

**Figure 7.** Illustration of the right SLF white matter bundles and their FA content in SBPp and SBPr patients of the New Haven data set. (**A**) Three-dimensional illustration of SLF tracts connecting the inferior parietal lobe with frontal operculum (left), inferior parietal lobe and middle frontal gyrus (middle), and superior parietal lobe and primary motor cortex (right). Upper two rows show SLF tracts connecting the two regions in 3D with FA values along those tracts shown in blue-to-red. The first row shows two representative SBPr patients and the second row shows two representative SBPp patients. (**B**) Plot of average FA ± SEM along the tracts depicted in A for 15 SBPr and 12 SBPp patients. The histogram plot shows the average number of tracts for these same patient groups. IPL, inferior parietal lobe; PFO, prefrontal operculum; MFG, middle frontal gyrus; SPL, superior parietal lobe; M1, primary motor cortex.

The online version of this article includes the following source data and figure supplement(s) for figure 7:

**Source data 1.** The complete list of the cortical and subcortical regions based on the Desikan-Killiany atlas (DKA) used to define the regions of interest (ROI) corresponding to the nodes in the structural connectivity analysis.

**Figure supplement 1.** Illustration of the left SLF white matter bundles and their FA content in SBPp and SBPr patients of the New Haven data set.

**Figure supplement 2.** Scatterplot of the testing AUC vs. the model validation AUC for all the machine learning models applied to the structural connectivity data.

best performing structural connectivity model was based on combining 40% of the New Haven data with 100% of the Chicago data during initial training and validation and testing on the remaining 60% of the New Haven data or on the Mannheim data. The average AUC (i.e. the average performance of the model) was 0.67±0.03 (mean ±std) using a support vector classifier (SVC) when classifying the remainder of the New Haven sample and 0.53±0.03 when classifying the Mannheim data (*Figure 7—figure supplement 2*).

## Discussion

The white matter properties of the right SLF were significantly different between SBPr and SBPp patients across three different sites: New Haven, Chicago, and Mannheim. Furthermore, SBPr patients showed larger FA values than pain-free controls in the right SLF in both the New Haven and Mannheim data sets. This suggests that this white matter property is a biomarker of resilience to pain chronicity where pain-free controls are composed of both high and low resilience individuals. Consistent with the concept of resilience, higher baseline fractional anisotropy in the right SLF predicted a greater percentage reduction in pain intensity at follow-up. A model based on whole-brain structural connectivity classified SBPp and SBPr patients across sites, although with a lower overall accuracy compared to the TBSS-based univariate approach. Despite the implication of the uncinate fasciculus in learning and memory, (*Von Der Heide et al., 2013*) which are important processes in the emergence of chronic pain, (*Timmers et al., 2019*) we did not detect any microstructural alterations in this tract. Alterations of the uncinate fasciculus were noted in the fiber complexity (*Bishop et al., 2018*) but not in fractional anisotropy (*Lieberman et al., 2014*) in CBP patients compared to pain-free controls. This suggests

that either FA is not the appropriate measure used for studying the uncinate fasciculus in relation to pain, or that this tract is not involved in chronic aversive learning.

The SLF is a large bundle of white matter association fibers connecting occipital, temporal, and lateral parietal lobes with the ipsilateral frontal lobe (*Thiebaut de Schotten et al., 2011*). In vivo studies in humans have anatomically subdivided the SLF into dorsal (SLF I), middle (SLF II), and ventral (SLF III) branches that run in parallel, (*Makris et al., 2005*) while the inclusion of the fourth component, the arcuate fasciculus, has been questioned by some authors due to its distinct anatomical trajectory that runs from the frontal to the temporal lobe (*Janelle et al., 2022*). The SLF is an essential anatomical substrate for major cognitive functions such as language (left SLF), memory, emotions, motor planning, and visuospatial processing (*Mesulam, 2000*). The right SLF is particularly important in visuospatial attention to the extra-personal space and the positioning of the body in the physical space (*Amemiya and Naito, 2016*). It connects frontal and parietal areas critical in the top-down control of attention during tasks involving any sensory modality (*Mesulam, 2000*). The SLF cluster found in our discovery set and validated in two other independent data sets spans all three branches of the SLF white matter tract. Within the significant cluster in the discovery data set, MD was significantly increased, while RD in the right SLF was significantly decreased in SBPr compared to SBPp patients. Higher RD values, indicative of demyelination, were previously observed in chronic musculoskeletal patients across several bundles, including the superior longitudinal fasciculus (*Lieberman et al., 2014*). Similarly, Mansour et al. found higher RD in SBPp compared to SBPr in the predictive FA cluster. While they noted decreased AD and increased MD in SBPp, suggestive of both demyelination and altered axonal tracts (*Mansour et al., 2013*), our results show increased MD and RD in SBPr with no AD differences between SBPp and SBPr, pointing to white matter changes primarily due to myelin disruption rather than axonal loss, or more complex processes. Further studies on tissue microstructure in chronic pain development are needed to elucidate these processes.

Studies in patients with brain lesions have supported the importance of the right SLF in top-down attention processing. For example, patients who underwent glioma resection with damages to the right medial superior and middle frontal gyrus, brain regions traversed by SLF I and II, showed persistent visuospatial cognitive dysfunction postoperatively (*Nakajima et al., 2017*). Similarly, patients with right prefrontal glioma with the resection cavity located in a region overlapping SLF I and SLF II had persistent spatial working memory deficits even in the absence of motor and language deficits (*Kinoshita et al., 2016*). Lesions within the SLF II have been linked to visuospatial neglect, (*Vallar et al., 2014*; *Thiebaut de Schotten et al., 2014*) supporting the role of this subcomponent in visuospatial awareness. Together the literature on the right SLF role in higher cognitive functions suggests, therefore, that resilience to chronic pain might be related to a top-down phenomenon involving visuospatial and body awareness.

Higher FA values in the right SLF have been associated with better performance in sustained attention in children (*Klarborg et al., 2013*) and people suffering from attention deficits or hyperactivity disorder (*Makris et al., 2008*; *Konrad et al., 2010*; *Wolfers et al., 2015*). After a stroke, patients who had higher baseline FA values within SLF II showed higher success rates in the visuomotor task after 4 weeks of learning compared to those with lower baseline FA values (*Buch et al., 2012*). Similarly, a recent study showed that SLF II underwent plastic changes after learning of tracking movement tasks requiring both top-down attentional processes and bottom-up somatosensory feedback to adjust one's own movement, and the degree of this plasticity predicted task-related success (*Shiao et al., 2022*). Since the constant adaptation of motor control in these tasks is dependent on spatial awareness and proprioception, the integrity of the SLF II appears crucial to their intact functioning.

Proprioception, which is a bottom-up somatic signal closely related to visuospatial processing, is critical for recognizing one's own body position and hence the awareness of the physical self. Changes at the higher-order level of proprioceptive processing can also affect body perception (*Goossens et al., 2019*). The right SLF connects a network of brain regions involved in proprioceptive awareness and allows us to perceive ourselves as separate entities from external animate and inanimate objects (*Naito et al., 2016*). Lesions of this white matter bundle and/or of the right parietal lobe are associated with hemispatial neglect of the left side of extra-personal space and, in extreme cases, of one's own body parts (*Mesulam, 2000*). Specifically, SLF II and SLF III subcomponents have been linked to dysfunctions in proprioception. The inferior frontoparietal network, connected by the SLF III, showed predominant right-hemispheric activation during both the visual self-face recognition and

limb proprioceptive illusion task, which suggests importance of this tract in self-awareness independent of the sensory modality and body parts involved (*Morita et al., 2017*). Similarly, changes in limb position in relation to posture activate the same network, hence suggesting that constant awareness of our body position and corrective feedback on our body schema is indeed dependent on this tract (*Amemiya and Naito, 2016*).

Although the findings on the direct association between impaired proprioception and CBP are inconclusive due to the variety of methods used to measure proprioceptive performance, (*Tong et al., 2017*; *Lin et al., 2019*; *Ghamkhar and Kahlaee, 2019*) there is evidence that the central processing of back-related proprioceptive signals is affected in chronic low back pain (*Goossens et al., 2019*; *Tsay et al., 2015*). Indeed, it has been shown that pain perception influences body representation, (*Schwoebel et al., 2001*) and body representation is altered in patients with chronic pain as investigated by psychophysical,(*Tsay et al., 2015*; *Gilpin et al., 2015*) and neuroimaging studies (*Goossens et al., 2019*; *Flor et al., 2006*; *Moseley and Flor, 2012*). Interestingly, in some cases disturbances in body representation occurred before the development of chronic pain (*Bultitude and Rafal, 2010*) and predicted decreased analgesic response to an exercise treatment, (*Tanaka et al., 2021*) suggesting a causal role of such a distortion in chronicity. While we did not collect proprioceptive and attention measures in the SBP patients, our results suggest that these measures could be useful to separate resilient and at-risk patients.

The asymmetry of the functions subserved by the SLF is notable as lesions to the right but not left SLF lead to hemispatial neglect on the left side, suggesting that vulnerabilities and strength in the neural substrate mediating attention and visuospatial processing contribute to the long-term risk or resilience to chronic pain. The observation that the structural properties of such a large association fiber network predict pain chronicity also suggests that risks and vulnerabilities to chronic pain may be determined by large, distributed frontoparietal networks on the right side of the brain. Interestingly, compared to pain-free controls, low back pain patients show functional connectivity alterations in the dorsal visual stream involved in visuospatial attention and subserved by the SLF tract, but not in the ventral visual stream (*Shen et al., 2019*). The results motivated us to explore different diffusion-based parameters, primarily focusing on structural connectivity. While our SVC machine learning-based model accurately classified SBPr and SBPp patients, it did not perform better than the univariate TBSS-based analysis and was worse than the latter approach in classifying the Mannheim patients. This discrepancy is most likely because whole-brain structural connectivity approaches such as PSC (*Zhang et al., 2018*) are more sensitive to signal-to-noise ratio and site-related differences in data acquisition parameters than TBSS such as the number of directions in the diffusion data acquisition protocol. Another explanation is that structural connectivity-based classifiers are sensitive to the heterogeneity of patients across sites because the eligibility criteria used were similar between New Haven and Chicago but different from those in Mannheim.

Our results along with previous studies suggest that resilience to chronicity of back pain may be related to the structural integrity (i.e. FA) of the right SLF. Like (*Mansour et al., 2013*), we observed that SBPr patients show larger FA values than the pain-free controls while the SBPp patients show similar FA values to the CBP patients suggesting that the pain-free population is made up of high and low-risk groups (*Figure 1*; *Figure 3*). Notably, though, the DTI-based FA results reported by Mansour et al. were on the left side of the brain. While larger datasets are required to explain this discrepancy, it does suggest that resilience to chronic pain might also be a widespread brain property involving other large-scale brain networks. CBP state has been associated with white matter changes across several tracts, (*Lieberman et al., 2014*) but here we show that the SLF tract is particularly important in the transition phase of subacute to CBP state, predicting long-term chronicity in a lateralized, right-dominant manner. As pain turns chronic, it is likely to progressively involve other neural pathways, indicating changes within broader neural networks. In fact, neural signatures of sustained and chronic pain are predominately observed in somatomotor, frontoparietal, and dorsal attention networks, (*Lee et al., 2021*) which corresponds to the microstructural changes in tracts found in the chronic state. The right SLF, however, shows changes already at half a year into chronicity, remains stable at one year, and could therefore serve as a potential biomarker to address the need for early detection of risk.

In task-based and resting state neuroimaging studies the brain activity in the frontoparietal area was not typically observed as a risk factor for persistent pain. Instead, functional connectivity between the medial prefrontal cortex and nucleus accumbens, (*Baliki et al., 2012*) as well as the processing of

reward signals in the same pathway (*Löffler et al., 2022*) were shown to be predictive of back pain chronicity. Investigations into structural properties showed that the risk for the development of CBP is related to a smaller nucleus accumbens and hippocampi (*Baliki et al., 2012*; *Vachon-Presseau et al., 2016*; *Makary et al., 2020*) in the SBPp patients relative to the other groups. Conversely, larger amygdala volume seems to be a protective factor against chronicity, as it was greater in SBPr than in both the persistent pain group and the pain-free group (*Vachon-Presseau et al., 2016*). In the same vein, we observed greater FA values within SLF in SBPr not only compared to SBPp but also to healthy controls. The involvement of the frontoparietal area can thus be regarded as part of structural 'resilience circuitry'.

Despite different time frames for the follow-up, initial (baseline) pain intensities (and accordingly different criteria for subgrouping SBP into SBPr/SBPp), population, sites, scanners, and pain questionnaires/screening used, we successfully validated results across three different sites. Moreover, we carefully addressed the potential confounding effects of head displacement, which can lead to either positive or negative bias (*Ling et al., 2012*) by accounting for it in the analysis. This points towards the robustness of the integrity of the SLF as a biomarker of resilience to CBP, with a potential for clinical translation.

## Limitations

Our results are based on heterogeneous samples, heterogeneous pain measures, different criteria for recovery, and different scanners across sites. In addition, at the time of analysis, we had 'access' to all the data, which may lead to bias in model training and development. We believe that the data presented here is nevertheless robust since multisite validated but needs replication. Additionally, we followed standard procedures for machine learning where we never mix the training and testing sets. The models were trained on the training data with parameters selected based on cross-validation within the training data. Therefore, no models have ever seen the test data set. The model performances we reported reflect the prognostic accuracy of our model. Even though our model performance is average-to-good, which currently limits its usefulness for clinical translation, we believe that future models would further improve accuracy by using larger homogenous sample sizes and uniform acquisition sequences. Future studies could validate our results with increased sample sizes and using the same criteria across sites. In addition, our studies did not evaluate functions subserved by the right SLF such as proprioception or other types of visuo-spatial tasks. We believe that the results strongly support the future assessment of such cognitive functions in the study of risk and resilience to chronic pain.

## Conclusions and future directions

We have identified a brain white matter biomarker of resilience to CBP and validated it in multiple independent cohorts at different sites. This biomarker is easy to obtain (~10 min of scanning time) and could open the door for translation into clinical practice, as future models on diffusion data are likely to improve accuracy by obtaining the data from larger sample sizes and using the same acquisition sequences. Although chronic pain may eventually affect other neural networks, the microstructural changes in the right SLF tract are evident early in the course of the illness and remain stable as pain progresses. Future studies should investigate how this brain structural predisposition to CBP may impact brain function, information processing, and neural networks. This could lead to potential neural targets for early interventions such as neurofeedback (*Bucolo et al., 2022*) or brain stimulation (*Kandić et al., 2021*). In addition, cognitive and behavioral processes associated with the right SLF, such as proprioception and attentional functions, should be examined in subacute stages, as targeting these processes could add to the effective prevention of chronicity. Integrating findings from studies that used questionnaire-based tools and showed remarkable predictive power, (*Tanguay-Sabourin et al., 2023*) with neurobiological measures that can offer mechanistic insights into chronic pain development, could enhance predictive power in CBP prognostic modeling.

To establish the clinical usefulness of a biomarker, it should be tested in various populations, settings, and contexts, and ideally be cost-effective and simple to implement (*Davis et al., 2020*). In this regard, we have introduced a promising biomarker found in brain white matter, which has considerable potential for clinical application in the prevention and treatment of CBP.

# Materials and methods

**Key resources table**

| Reagent type (species) or resource | Designation | Source or reference | Identifiers | Additional information |
|---|---|---|---|---|
| Software, algorithm | FSL 6.0.0 | Oxford Centre for Functional MRI of the Brain | RRID:SCR_002823 | |
| Software, algorithm | R 4.3.2 | R Foundation for Statistical Computing | RRID:SCR_001905 | |
| Software, algorithm | Python 3.11 | Python Software Foundation | RRID:SCR_008394 | |
| Software, algorithm | Original analysis code | This paper | https://www.openpain.org/ under WMPainResiliencePathway dataset | See Materials and methods, section 'Estimation of structural connectivity' |

## Data pool

### New Haven (Discovery) data set

We recruited individuals in the New Haven, Connecticut area. Subjects were recruited through flyers and internet advertisements. All participants gave written informed consent to participate in the study. The study was approved by the Yale University Institutional Review Board.

Brain diffusion data were collected at baseline from 27 (12 females) subacute low-back pain patients (SBP, pain duration between 6–12 weeks), 29 (16 females) patients with chronic low-back pain (CLBP), and 28 (12 females) healthy controls (HC). One patient was excluded because of excessive head motion defined as >3 SD from the mean Euclidian distance of either translational or rotational displacement during the MRI scanning session and one HC subject was excluded because parts of the brain were outside the field of view.

Subjects were briefly screened at first to check *Hoy et al., 2012* the location of the low-back pain, (*Lo et al., 2021*) if they were otherwise healthy, (*Ferreira et al., 2023*) non-smokers, and (*Deyo et al., 2015*) pain duration (between 6 and 12 weeks for SBP and more than one year for CLBP). If they passed this initial brief screen a more detailed screen was conducted where we assessed complete

**Table 1.** New Haven sample characteristics.

| | HC (N=28) | CLBP (N=28) | SBPr (N=16) | SBPp (N=12) | t(df)*, p-value | Missing |
|---|---|---|---|---|---|---|
| Age (years) | 30.1 (10.1) | 30.7 (11.9) | 30.8 (8.8) | 38.0 (12.5) | +1.80 (26), 0.08 | 0/0/0/0 |
| Gender (m/f) | 16/12 | 12/16 | 11/5 | 7/5 | +0.32 (1), 0.57[†] | 0/0/0/0 |
| BMI (Kg/m2) | 24.1 (3.6) | 24.2 (3.7) | 25.5 (5.2) | 26.2 (4.5) | +0.35 (26), 0.73 | 0/0/0/0 |
| Delta pain severity: absolute | NA | NA | - 25.4 (15.4) | 8.0 (17.2) | **+5.0 (26),$<10^{-4‡}$** | NA/NA/0/0 |
| Delta pain severity: percentage | NA | NA | - 66.3 (26.9) | 39.1 (68.1) | **+5.7 (26),$<10^{-5‡}$** | NA/NA/0/0 |
| Pain Duration | NA | 5.3 (4.7) | 8.6 (3.6) | 10.9 (3.1) | +1.87 (26), 0.08 | 0/0/0/0 |
| Pain Intensity | NA | 4.5 (2.0) | 36.7 (18.8) | 33.7 (15.9) | - 0.45 (26), 0.66 | NA/0/0/0 |
| BDI | 2.6 (3.3) | 6.4 (6.0) | 7.4 (4.8) | 3.1 (3.8) | - 2.65 (26), 0.02[‡] | 0/0/0/0 |
| BAI | 3.4 (5.7) | 7.6 (8.2) | 6.4 (6.8) | 4.8 (2.8) | - 0.75 (26), 0.46 | 0/0/0/0 |
| MPQ | NA | 10.6 (4.6) | 9.2 (5.2) | 9.1 (4.2) | - 0.08 (26), 0.93 | NA/0/0/0 |
| PCS | NA | 15.2 (9.8) | 12.8 (11.4) | 11.3 (8.1) | - 0.40 (26),0.69 | NA/0/1/0 |

Abbreviations: BMI, body mass index; BDI, Beck's Depression Index; BAI, Beck's Anxiety Index; MPQ, McGill Pain Questionnaire; PCS, Pain Catastrophizing Scale. Values show the group mean and standard deviation in parenthesis.

*t-score (degrees of freedom).

[†]Chi-square test.

[‡]$p < 0.05$.

medical and psychiatric history. To be included in the study, SBP subjects needed to meet the criteria of having a new-onset 6–12 weeks low back pain at an intensity more than 20/100 on the visual analogue scale (VAS) and report being pain-free in the year prior to the onset of back pain. CLBP patients had to have a pain duration of at least 1 year and a pain intensity of more than 30/100. Both SBP and CLBP participants had to *Hoy et al., 2012* fulfill the International Association for the Study of Pain criteria for back pain, (*Treede, 2018*; *Lo et al., 2021*) not be currently, or during the month prior to the study, on any opioid analgesics. Patients were included if their back pain was below the 12th thoracic vertebra with or without radiculopathy and was present on more days than not. SBP and CLBP diagnoses were confirmed based on history collected by an experienced clinician (P.G.). Healthy control subjects were screened likewise with, besides, the absence of any history of any pain of more than 6 weeks in duration. Participants had no history of mental disorders, chronic medical conditions (e.g. diabetes, coronary artery disease), or loss of consciousness.

The study consisted of two time points separated by approximately one year (baseline, 1 year follow-up). At each time point participants completed one testing session in the laboratory and one scanning session. Patients whose pain dropped by more than 30% (*Dworkin et al., 2005*) at follow-up were considered recovered, otherwise persistent. Of the SBP group, 12 patients were confirmed at follow-up as recovered subacute back pain patients (SBPr) and 16 as persistent subacute back pain patients (SBPp) and had diffusion tensor data collected. For demographic and clinical characteristics of the different groups see *Table 1*.

Magnetic resonance imaging. Participants underwent an anatomical T1-weighted scan and two consecutive 2.5-min-long diffusion tensor imaging (DTI) scans in the same session. A Siemens 3T Trio B magnet (SIEMENS Healthineers, Erlangen, Germany) equipped with a 32-channel head coil was used to acquire the images. The 3D magnetization prepared rapid gradient echo (MPRAGE) T1-weighted acquisition sequence was as follows: repetition time/echo time (TR/TE) = 1900/2.52ms, flip angle = 9°, matrix size = 256 x 256, number of slices = 176, image resolution = = 1 × 1×1 mm$^3$. The diffusion-weighted images were acquired using spin-echo echo planar imaging (SE-EPI) sequence using the following scan parameters: TR/TE = 2200/84.0ms, flip angle = 90°, matrix size = 110 × 110×64, multi-band acceleration factor = 4, image resolution = 2 × 2×2 mm$^3$. Diffusion gradients were applied along 64 directions with a b-value of 1000 s/mm$^2$. For each set of DTI data, one volume with no diffusion weighing (i.e. b=0 s/mm$^2$) was acquired at the beginning of the scan.

## Mannheim data set

Participants were recruited through advertisements in local newspapers and the website of the Central Institute of Mental Health in Mannheim, and patients were recruited additionally through the outpatient pain clinic of the Institute of Cognitive and Clinical Neuroscience, general practitioners, and physiotherapy practices. We determined subjects' eligibility via a telephone screening form, which comprised questions about MRI contraindications, medication intake, current and previous drug/alcohol use, co-morbid medical and psychological conditions, and pain frequency and severity. All participants had to be between 18 and 70 years old to be eligible for study entry. Healthy controls had to be pain-free; patients with pain (SBP and CBP) had either low and/or upper back pain. To be included in the study, SBP participants had to have a current back pain episode of 7–12 weeks duration. Patients with a current back pain episode and additional back pain episodes in their history were included as well if the episodes never exceeded 12 weeks per year. The inclusion criteria for the CBP group were a history of back pain longer than 6 months and a current back pain episode of more than 100 days. Participants were excluded if they reported any neurological disorder, psychotic episodes, current substance abuse, a major illness, contraindication for MRI, or another painful condition as the main pain problem (see also for comorbid mental disorders). In addition, all subjects completed the German versions of the Chronic Pain Grade (CPG; *Von Korff et al., 1992*), the Örebro Musculoskeletal Pain Questionnaire (OMPQ yellow flag; *Langenfeld et al., 2018*), the Hospital Anxiety and Depression Scale (HADS; *Snaith et al., 1995*), and the Perceived Stress Scale (PSS; *Reis et al., 2019*). Regular medication use was reported as follows: NSAIDs (6 SBP, 3 CBP), statins (2 CBP), angiotensin receptor blockers (2 SBP), proton-pump inhibitors (1 SBP), antihistamines (1 CBP). Occasional use was reported for NSAIDs (1 HC, 2 SBP, 3 CBP), angiotensin receptor blockers (1 CBP), antihistamines (1 HC, 1 CBP), and benzodiazepines (1 CBP). Past use of NSAIDs was reported for 1 HC, 22 SBP, 14 CBP; of benzodiazepines for 1 SBP, 2 CBP; of ACE inhibitors for 1 CBP, and of Cannabinoids for 1 CBP.

Brain diffusion data were collected at baseline from 64 patients with SBP, 24 patients with CBP, and 24 healthy controls (HC). HC and patients with CBP were matched for age and gender. Two HC, three patients with CBP, and nine patients with SBP were excluded from the analysis due to excessive head motion defined as >3 SD from the mean Euclidian distance of either translational or rotational displacement during the MRI scanning session. Additionally, two SBP patients' diffusion images failed manual quality control checks due to obvious artifacts. Six patients with SBP were excluded from the analysis because they did not have interim data at follow-up. One patient with SBP was also excluded

**Table 2.** Mannheim sample characteristics.

| | HC (N=22) | CBP (N=21) | SBPr (N=28) | SBPp (N=18) | t(df)*, p-value | Missing |
|---|---|---|---|---|---|---|
| Age in years | 36.9 (14.4) | 40.0 (16.0) | 32.8 (12.0) | 32.1 (11.2) | +0.19 (38.2), 0.85 | 0/0/0/0 |
| Gender (m/f) | 12/10 | 11/10 | 8/20 | 6/12 | +0.0002(1),0.99* | 0/0/0/0 |
| Number of days with pain during last year | NA | 247 (84.0) | 74.8 (43.2) | 72.4 (34.9) | +0.2 (41.5,2), 0.84 | NA/1/0/0 |
| Delta pain severity (FU-BL): absolute | NA | NA | –1.90 (1.18) | 0.19 (0.71) | –7.47 (43.8),<$10^{-9}$[†] | NA/NA/0/0 |
| Delta pain severity (FU-BL): percentage | NA | NA | –50.9 (27.3) | 8.73 (26.0) | –7.45 (37.7),<$10^{-9}$[†] | NA/NA/0/0 |
| Pain severity (MPI) | NA | 4.92 (1.38) | 3.80 (1.43) | 3.78 (1.54) | +0.06 (34.5), 0.95 | NA/1/0/0 |
| Interference (MPI) | NA | 2.43 (1.23) | 1.48 (1.01) | 1.49 (1.00) | –0.03 (32), 0.97 | 3/0/2/2 |
| Negative mood (MPI) | NA | 2.83 (1.07) | 2.40 (1.08) | 2.71 (1.15) | –0.87 (30.5), 0.39 | 3/0/2/2 |
| Life control (MPI) | NA | 3.97 (0.971) | 4.05 (1.24) | 3.54 (1.23) | +1.3 (32), 0.2 | 3/0/2/2 |
| Support (MPI) | NA | 2.60 (1.85) | 1.85 (1.28) | 1.56 (1.41) | +0.66 (29.4), 0.52 | 3/0/2/2 |
| ÖMPQ | NA | 78.1 (19.5) | 63.6 (21.1) | 69.7 (16.1) | –1.06 (37.9), 0.3 | 12/1/2/2 |
| CPG[a] | NA | 1.00 [0, 6.00] | 0 [0, 4.00] | 0 [0, 6.00] | –0.65 (25.9), 0.52 | 9/0/0/2 |
| Active coping (PRSS) | NA | 3.30 (0.74) | 2.94 (0.96) | 3.14 (0.66) | –0.76 (36.6), 0.45 | NA/6/4/3 |
| Catastrophizing (PRSS) | NA | 1.35 (0.87) | 1.09 (0.73) | 1.10 (0.76) | –0.02 (28.9), 0.98 | NA/6/4/3 |
| Anxiety (HADS) | 4.37 (2.39) | 8.24 (4.47) | 7.27 (4.63) | 7.47 (3.60) | –0.15 (35.3), 0.88 | 3/0/2/3 |
| Depression (HADS) | 6.21 (6.12) | 6.00 (4.40) | 4.42 (4.23) | 4.60 (2.85) | –0.15 (37.9), 0.87 | 3/0/2/3 |
| Perceived stress (PSS) | 4.74 (5.00) | 11.0 (5.23) | 12.0 (5.25) | 11.4 (5.07) | +0.34 (32.7), 0.73 | 3/0/2/2 |

MPI, West Haven-Yale Multidimensional Pain Inventory; CPG, Chronic Pain Grade; ÖMPQ, Örebro Musculoskeletal Pain Questionnaire; PRSS, pain related self-statements; HADS, Hospital Anxiety and Depression Scale; HC, healthy control; CBP, chronic back pain; SBP, patients with subacute back pain; SBPp/r, patients with SBP with persistent pain or recovered pain after 6 months; FU, follow-up after 6 months; BL, baseline assessment; SD, standard deviation. Values show the group mean and standard deviation in parenthesis. [a]For the Chronic Pain Grade the median and interquartile range are shown.

Chi-square test.
*t-score (degrees of freedom).
[†] p < 0.05.

from the analysis as an outlier due to an extreme increase in pain severity from baseline to follow-up (466 percent change, M = 2300 = –15,06; hence the subject was >6 standard deviations from the sample mean). The final sample in the analysis comprised 22 HC, 21 patients with CBP, and 46 patients with SBP. Demographic and clinical characteristics of this data set, which was used for validation of our white matter biomarker, are presented in *Table 2*.

All participants gave written informed consent to be involved in the study and received 10€/hour for their participation. The study was approved by the Ethics Committee of the Medical Faculty of Mannheim, Heidelberg University, and was conducted in accordance with the declaration of Helsinki in its most recent form.

Magnetic resonance imaging. A 3 Tesla Tim TRIO whole body scanner (SIEMENS Healthineers, Erlangen, Germany), equipped with a 12-channel head coil was used to acquire the images. Shimming of the scanner was done to account for maximum magnetic field homogeneity. Participants underwent an anatomical T1-weighted MPRAGE imaging scan with the following sequence: TR/TE = 2300 / 2.98ms, flip angle = 9°, matrix size = 240 x 256, number of slices = 192, image resolution = 1 x 1 x 1 mm3. The diffusion weighted images were acquired using SE-EPI sequence with the scan parameters as follows: TR/TE = 7400/85ms, matrix size = 220 x 256 x 30, GRAPPA = 2, image resolution = 2 x 2x 2 mm3. Diffusion gradients were applied along 30 directions using a b-value of 1000 s/mm2. One volume with no diffusion weighing was acquired at the beginning of the scan.

Clinical assessments. Patients with SBP were included in the study at baseline as one group, and their pain severity was assessed at two time points (baseline, 6-month follow-up). Change in pain severity (PS) was assessed using the percentage change in the Pain Severity scale of the German version of the West Haven-Yale Multidimensional Pain Inventory (*Flor et al., 1990*) from baseline assessment to the follow-up screening after 6 months using the following formula : $\Delta PS = \frac{PSfollow-up - PSbaseline}{PSbaseline} \times 100$. Based on the pain severity percentage change, the SBP sample was divided into recovered SBP (SBPr, N=28), whose pain dropped by more than 20% at follow-up and persisting SBP (SBPp, N=18) (≥; *Baliki et al., 2012*). Comorbid mental disorders are presented in .

**Table 3.** Chicago (Open Pain) sample characteristics.

| | SBPr (N=23) | SBPp (N=35) | t(df)*, p-value | SBPr (N=28) | SBPp (N=34) | t(df)*, p-value | **Missing** |
|---|---|---|---|---|---|---|---|
| Age (years) | 41.7 (12.0) | 43.6 (9.3) | +0.7 (56), 0.48 | 43.7 (11.5) | 45.3 (9.6) | +0.61 (60), 0.54 | 0/0/0/0 |
| Gender (m/f) | 12/9 | 16/19 | 1.6 (1),0.2[†] | 15/13 | 17/17 | 0.08 (1), 0.78[†] | 0/0/0/0 |
| Delta pain severity: absolute | - 40.6 (20.8) | –3.4 (15.6) | +7.8 (56),<10⁻⁶ [‡] | - 43.1 (20.6) | - 3.2 (15.6) | **+8.7 (60),<10⁻⁶[‡]** | 0/0/0/0 |
| Delta pain severity: percentage | - 68.9 (26.2) | –1.6 (31.9) | +8.4 (56),<10⁻⁶ [‡] | - 69.7 (25.9) | 0.3 (31.6) | **+9.3 (60),<10⁻⁶[‡]** | 0/0/0/0 |
| Pain Duration (weeks) | 9.9 (4.1) | 8.4 (4.3) | - 1.3 (55), 0.18 | 67.8 (5.6) | 65.2 (5.7) | - 1.8 (60), 0.07 | 0/1/0/0 |
| Pain Intensity | 58.0 (15.2) | 67.7 (17.2) | +2.3 (56),0.03[‡] | 18.0 (16.8) | 65.7 (15.8) | **+11.5 (60),<10⁻⁶[‡]** | 0/0/0/0 |
| BDI | 5.7 (5.2) | 7.3 (4.6) | +1.0 (39), 0.31 | 6.3 (6.6) | 16.0 (9.5) | +1.4 (42), 0.16 | 7/10/10/8 |
| MPQ | 10.9 (4.5) | 18.2 (17.9) | +1.9 (54), 0.06 | 13.4 (26.3) | 16.0 (9.5) | +4.3 (56), <10⁻⁴ [‡] | 0/2/3/1 |

* t-score (degrees of freedom).
[†]Chi-square test.
[‡] p < 0.05.

## Chicago data set (Open Pain)

The data set obtained through the https://www.openpain.org/ online database (collected in Chicago, from now on, we refer to this data set as 'Chicago data set') had 58 SBP patients (28 females) with a baseline visit (visit 1 on OpenPain, pain duration 6–12 weeks) and 60 SBP patients (29 females) with a 1-year follow-up visit (visit 4 on OpenPain) on whom diffusion images were collected.

Patients were deemed recovered at 1-year follow-up based on the same criterion as the New Haven data (i.e. >30% drop in low-back pain intensity reported on the VAS). As such, we studied 35 SBPp and 23 SBPr at baseline, and 33 SBPp and 27 SBPr at follow-up. Demographic and clinical characteristics of this data set, which was used for validation of our white matter biomarker, are presented in *Table 3*.

Magnetic resonance imaging. Part of this data has been already published by *Mansour et al., 2013*. Therefore, we will briefly describe acquisition parameters. MPRAGE type T1-anatomical brain images were acquired with a 3T Siemens Trio whole-body scanner with echo-planar imaging capability using the standard radio-frequency head coil with the following parameters: TR/TE = 2500/3.36ms, flip angle = 9°, matrix size = 256 × 256; number of slices = 160, image resolution = 1 × 1×1 mm3. DTI images were acquired on the same day using SE-EPI with TR/TE = 9000/83, flip angle = 90 degrees, in-plane matrix resolution = 112 × 130; 73 slices; image resolution = 2 × 2×2 mm3. Images had an isotropic distribution along 60 directions using a b value of 1000 s/mm2. For each set of diffusion-tensor data, 8 volumes with no diffusion weighting were acquired at equidistant points throughout the acquisition.

## Preprocessing of DTI data

Preprocessing of all data sets was performed employing the same procedures and the FMRIB diffusion toolbox (FDT) running on FSL 6.0.0 (*Smith et al., 2004*) First, diffusion-weighted data were visually inspected for obvious artifacts or missing parts of the brain. Next, the data were corrected for eddy currents and head motion by employing affine registration to the no diffusion volume using eddy_openmp from the FSL toolbox. Eddy current corrects for image distortions due to susceptibility-induced distortions and eddy currents in the gradient coils (*Andersson and Sotiropoulos, 2016*). We do note, however, that as we did not acquire data in the phase-opposite direction, the susceptibility-induced distortions may not be fully corrected. Brain images were then skull stripped and a diffusion tensor model was fit, using FMRIB Diffusion Toolbox (FDT) part of FSL (*Behrens et al., 2003*), at each voxel to calculate the fractional anisotropy (*Basser et al., 1994*; *Smith, 2002*). FA reflects the degree of water diffusion within a voxel with values ranging between 0 and 1 where large values indicate directional dependence of Brownian motion due to white matter tracts and small values indicate more isotropic diffusion and less directionality (*Beaulieu, 2002*).

## Harmonization of DTI data using neuroCombat

Because the 3 data sets originated from different sites using different MR data acquisition parameters and slightly different recruitment criteria, we applied neuroCombat (*Fortin et al., 2017*) running in Python 3.11 to correct for site effects and then repeated the TBSS analysis shown in *Figure 1* and the validation analyses shown in *Figures 5 and 6*. First, the FA maps derived using FDT toolbox were pooled into one TBSS analysis where registration to a standard template FA template (FMRIB58_FA_1 mm.nii.gz part of FSL) was performed. Next, neuroCombat was applied to the FA maps as implemented in Python with batch (i.e., site) effect modeled with a vector containing 1 for New Haven, 2 for Chicago, and 3 for Mannheim originating maps respectively. The harmonized maps were then skeletonized to allow for TBSS.

## Tract-Based Spatial Statistics

Voxel-wise statistical analysis of FA was carried out using Tract-Based Spatial Statistics (TBSS) (*Smith et al., 2006*) part of the FSL (*Smith et al., 2004*). All subjects' FA data were then aligned into a common space (MNI standard 1 mm brain) using the nonlinear registration tool FNIRT, (*Andersson and Smith, 2007a*; *Andersson and Smith, 2007b*) which uses a b-spline representation of the registration warp field (*Rueckert et al., 1999*). Next, the mean FA image was created and thinned to create a mean FA skeleton, which represents the centers of all tracts common to the groups. Each subject's aligned FA data was then projected onto this skeleton and the resulting data fed into voxel-wise

cross-subject statistics. Groups (i.e. SBPr and SBPp) were compared using unpaired t-test corrected for age, gender, and motion parameters. Head displacement was estimated by eddy current correction to extract the magnitude of translations and rotations. Overall head motion was then calculated as the Euclidian distance from head translations and rotations for each subject, and these measures were Z-transformed before they were entered into the design matrix as nuisance variables. The statistical significance of TBSS-based testing was determined using a permutation-based inference (*Winkler et al., 2014*) where the null distribution is built using 10,000 random permutations of the groups. Significance was set at p<0.05, and significant clusters were identified using threshold-free cluster enhancement (*Smith and Nichols, 2009*).

## Statistical analysis

A mask formed from the significant cluster for the SBPr >SBPp contrast in the New Haven data was used to extract FA values from the Mannheim and the OpenPain data. The FA values were first corrected for confounders and then used to build a receiver operating curve (ROC) to assess classification accuracy (recovered and persistent pain patients as binary classes) based on the brain white matter data. The statistical significance of the area under the ROC (AUC) was tested against 10,000 random permutations of the group labels to generate a random distribution of the AUC values. Additionally, we tested if FA values predict pain percentage change in a dimensional approach using multiple linear regression with the FA data entered as a predictor, and pain percentage change as an outcome. In both, classification and linear regression analysis, age, gender, and motion parameters (translation and rotation) were entered as covariates of no interest. All statistical analysis were done in R 4.3.2.

## Estimation of structural connectivity

Structural connectivity was estimated from the diffusion tensor data using a population-based structural connectome (PSC) detailed in a previous publication (*Zhang et al., 2018*) PSC can utilize the geometric information of streamlines, including shape, size, and location for a better parcellation-based connectome analysis. It, therefore, preserves the geometric information, which is crucial for quantifying brain connectivity and understanding variation across subjects. We have previously shown that the PSC pipeline is robust and reproducible across large data sets (*Zhang et al., 2018*). PSC output uses the Desikan-Killiany atlas (DKA; *Desikan et al., 2006*) of cortical and sub-cortical regions of interest (ROI). The DKA parcellation comprises 68 cortical surface regions (34 nodes per hemisphere) and 19 subcortical regions. The complete list of ROIs is provided in the *Figure 7—source data 1*. PSC leverages a reproducible probabilistic tractography algorithm (*Maier-Hein et al., 2017*) to create whole-brain tractography data, integrating anatomical details from high-resolution T1 images to minimize bias in the tractography. We utilized DKA (*Desikan et al., 2006*) to define the ROIs corresponding to the nodes in the structural connectome. For each pair of ROIs, we extracted the streamlines connecting them by following these steps: (1) dilating each gray matter ROI to include a

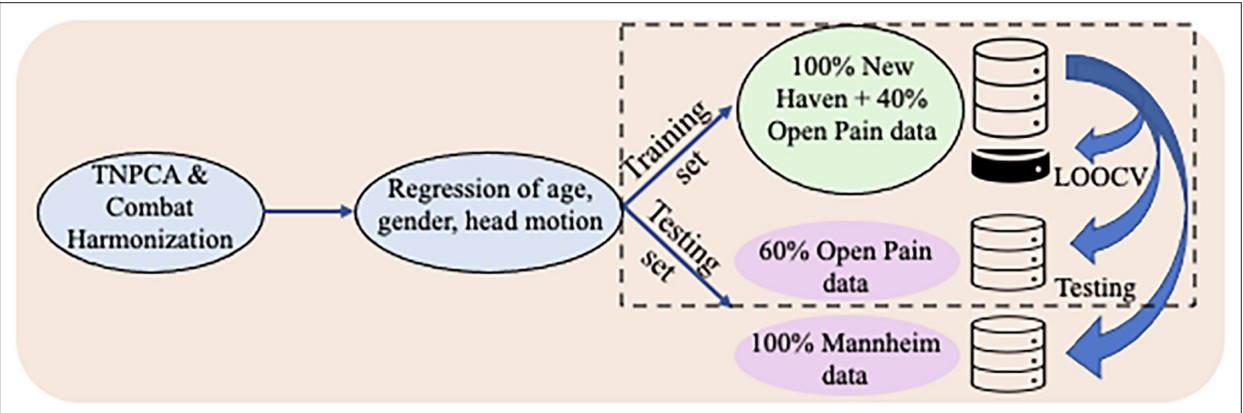

**Figure 8.** Schematic representation of the sequential steps performed in structural connectivity. From left to right: data preparation, correction for confounders, and machine learning model building and testing. The dashed rectangle indicates that the combination of the data during model training was bootstrapped 50 times and validation and testing were repeated accordingly.

**Table 4.** Summary of the type of data combinations expressed in % of subjects from each site used to build and test the brain connectivity-based machine learning models to classify recovered and persistent SBP patients.

| Training & Validation* | | Testing‡ | | |
| --- | --- | --- | --- | --- |
| New Haven | Chicago | New Haven | Chicago | Mannheim |
| 100% | 0% | 0% | 100% | 100% |
| 100% | 40% | 0% | 40% | 100% |
| 0% | 100% | 100% | 0% | 100% |
| 40% | 100% | 60% | 0% | 100% |

*The combination of data sets was bootstrapped 50 times and the training and testing was repeated accordingly.

small portion of white matter regions, (2) segmenting streamlines connecting multiple ROIs to extract the correct and complete pathway, and (3) removing apparent outlier streamlines. Due to its widespread use in brain imaging studies (*Chiang et al., 2011*; *Zhao et al., 2021*), we examined the mean fractional anisotropy (FA) value along streamlines and the count of streamlines in this work. The output we used includes fiber count, fiber length, and fiber volume shared between the ROIs in addition to measures of fractional anisotropy and mean diffusivity.

## Machine learning and cross-validation

The processing steps applied to the PSC output are summarized in *Figure 8* and were published previously (*Zhang et al., 2018*).

We used fiber count and total connected surface area connecting each pair of regions as the input features, which were 87x87 connectivity matrices of the DKA ROIs; the latter two features were chosen because of their demonstrated reliability in prior work (*Zhang et al., 2018*). Dimensions were reduced using tensor principal component analysis (TNPCA) with hyperparameter K=87. Next, because data originated from different sites (New Haven, Chicago, and Mannheim), we applied the ComBat harmonization (*Fortin et al., 2017*) to account for the effect of different sites on the data. Linear regression was used next to correct the data for age, gender, and head translation and rotation magnitudes. We used logistic regression, linear support vector classification (SVC), and random forest as our regression methods with leave-one-out cross-validation (LOOCV) and the number of components as a hyperparameter. Hence, the number of PCs included in the model was randomly chosen from a sequence of 22 numbers starting with 1 with an increment of 4 and a maximum of 87. The training of our regression models was performed using different combinations of data originating from New Haven and the Chicago database (as outlined in *Table 4*) and tested on the remainder of the data from these two sites or on the data originating from Mannheim.

The combination of the data during model training was bootstrapped 50 times and testing and validation were repeated accordingly. We chose to train the model only on the data originating from New Haven and Chicago because the eligibility criteria were quite similar between the two sets but different from those used to recruit SBP patients at Mannheim. The average performance of our classifier was based on the models that achieved a validation area under (AUC) the receiver operating characteristic curve (ROC) of ≥ 0.75 during training. Of note, such models cannot tell us the features that are important in classifying the groups. Hence, our model is considered a black-box predictive model like neural networks.

## Acknowledgements

The New Haven study was funded by the National Institute on Drug Abuse grant number K08DA037525. The Mannheim study was supported by a grant from the Deutsche Forschungsgemeinschaft (SFB1158/B03 to FN and HF) and it *has been registered on the "German Clinical Trials Register" the registration ID is DRKS00008835*. The access website is as follows: https://www.drks.

de/. The OPP project (Principal Investigator: A Vania Apkarian, Ph.D. at Northwestern University) is supported by the National Institute of Neurological Disorders and Stroke (NINDS) and National Institute on Drug Abuse (NIDA).

## Additional information

### Funding

| Funder | Grant reference number | Author |
|---|---|---|
| Deutsche Forschungsgemeinschaft | DRKS00008835 | Frauke Nees Herta Flor |
| National Institute on Drug Abuse | K08DA037525 | Paul Geha |

The funders had no role in study design, data collection and interpretation, or the decision to submit the work for publication.

### Author contributions

Mina Mišić, Conceptualization, Data curation, Formal analysis, Methodology, Project administration, Validation, Visualization, Writing – original draft, Investigation; Noah Lee, Conceptualization, Data curation, Formal analysis, Methodology, Validation, Visualization, Writing – review and editing; Francesca Zidda, Conceptualization, Project administration, Writing – review and editing; Kyungjin Sohn, Zhengwu Zhang, Software, Writing – review and editing; Katrin Usai, Martin Löffler, Conceptualization, Investigation, Writing – review and editing; Md Nasir Uddin, Arsalan Farooqi, Giovanni Schifitto, Writing – review and editing; Frauke Nees, Paul Geha, Herta Flor, Supervision, Funding acquisition, Writing – original draft, Writing – review and editing

### Author ORCIDs

Mina Mišić ⓘ https://orcid.org/0000-0003-1754-8681
Francesca Zidda ⓘ https://orcid.org/0000-0003-2507-0542
Katrin Usai ⓘ https://orcid.org/0000-0002-8534-3373
Zhengwu Zhang ⓘ https://orcid.org/0000-0002-9047-8838
Paul Geha ⓘ https://orcid.org/0000-0002-0537-7216

### Ethics

registration DRKS00008835.
All participants gave written informed consent to be involved in the study and permission to publish was obtained. The New Haven study was approved by the Yale University Institutional Review Board and Mannheim study was approved by the Ethics Committee of the Medical Faculty of Mannheim, Heidelberg University.

Reviewer #1 (Public review): https://doi.org/10.7554/eLife.96312.3.sa1
Reviewer #2 (Public review): https://doi.org/10.7554/eLife.96312.3.sa2
Reviewer #3 (Public review): https://doi.org/10.7554/eLife.96312.3.sa3
Author response https://doi.org/10.7554/eLife.96312.3.sa4

## Additional files

### Supplementary files

• MDAR checklist

• Supplementary file 1. Comorbid mental disorders assessment and characteristics – Mannheim data set. A trained psychologist interviewed all participants to assess comorbid mental disorders using the German version of the Structured Clinical Interviews (SCID I) for the Diagnostic and Statistical Manual of Mental Disorders (DSM IV)(*Wittchen, 1997*). List of all reported diagnoses across groups is provided in *Supplementary file 1—Table 1*.

## Data availability

The data set obtained from Chicago is from the OpenPain Project (OPP) publicly available at: https://www.openpain.org/ (this is an external dataset and a data usage agreement is required before access to the data). New Haven data and connectivity analysis scripts are likewise shared on OpenPain Project repository (WMPainResiliencePathways Database). The informed consent for Mannheim study does not give us permission to transfer the data in question to a repository of a Third Party for the purpose of unrestricted distribution to the research community. EU institutes can request the data directly from the author (mina.misic@zi-mannheim.de) and the Central Institute of Mental Health (CIMH) will provide data under a data access / transfer agreement. The consent does not cover transfers from CIMH to institutes outside the EU (as less strict data privacy rules may apply and the study participants were not informed about this). If CIMH receives a data requests from a non-EU organization, the CIMH will need to perform and document a process of weighing up competing interests (a so callled "Güter-Interessen-Abwägung") and only after an internal review board (IRB) approved the result of it and the requester signs the Standard Contracual Clauses (SCC), the data could be transferred in an anonymized form to ensure participant privacy and to comply with legal requirements to the requestor/organization outside the EU. In addition to the request, both - EU and non-EU requestors, would need to provide the project proposal for the intended use of the data and the approval of their ethics committee. Data cannot be used for commercial research.

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
