## [Editor Report · eLife Assessment]

This **valuable** study provides **convincing** evidence that white matter diffusion imaging of the right superior longitudinal fasciculus might help to develop a predictive biomarker of chronic back pain chronicity. The results are based on a discovery-replication approach with different cohorts, but the sample size is limited. The findings will interest researchers interested in the brain mechanisms of chronic pain and in developing brain-based biomarkers of chronic pain.

---

## [Referee Report · Reviewer #1 (Public review)]

Summary:

In this paper, Misic et al showed that white matter properties can be used to classify subacute back pain patients that will develop persisting pain.

Strengths:

Compared to most previous papers studying associations between white matter properties and chronic pain, the strength of the method is to perform a prediction in unseen data. Another strength of the paper is the use of three different cohorts. This is an interesting paper that provides a valuable contribution to the field.

Weaknesses:

The main weakness of this study is the sample size. It remains small despite having 3 cohorts. This is problematic because results are often overfitted in such a small sample size brain imaging study, especially when all the data are available to the authors at the time of training the model (Poldrack et al., Scanning the horizon: towards transparent and reproducible neuroimaging research, Nature Reviews in Neuroscience 2017). Thus, having access to all the data, the authors have a high degree of flexibility in data analysis, as they can retrain their model any number of time until it generalizes across all three cohorts. In this case, the testing set could easily become part of the training making it difficult to assess the real performance, especially for small sample size studies.

Even if the performance was properly assessed their models show AUCs between 0.65-0.70, which is usually considered as poor, and most likely without potential clinical use. Despite this, their conclusion was: "This biomarker is easy to obtain (~10 min 18 of scanning time) and opens the door for translation into clinical practice." One may ask who is really willing to use an MRI signature with a relatively poor performance that can be outperformed by self-report questionnaires?

Overall, these criticisms are more about the wording sometimes use and the inference they made. I still think this is a very relevant contribution to the field. Showing predictive performance through cross validation and testing in multiple cohorts is not an easy task and this is a strong effort by the team. I strongly believe this approach is the right one and I believe the authors did a good job.

---

## [Referee Report · Reviewer #2 (Public review)]

The present study aims to investigate brain white matter predictors of back pain chronicity. To this end, a discovery cohort of 28 patients with subacute back pain (SBP) was studied using white matter diffusion imaging. The cohort was investigated at baseline and one-year follow-up when 16 patients had recovered (SBPr) and 12 had persistent back pain (SBPp). A comparison of baseline scans revealed that SBPr patients had higher fractional anisotropy values in the right superior longitudinal fasciculus (SLF) than SBPp patients and that FA values predicted changes in pain severity. Moreover, the FA values of SBPr patients were larger than those of healthy participants, suggesting a role of FA of the SLF in resilience to chronic pain. These findings were replicated in two other independent datasets. The authors conclude that the right SLF might be a robust predictive biomarker of CBP development with the potential for clinical translation.

Developing predictive biomarkers for pain chronicity is an interesting, timely, and potentially clinically relevant topic. The paradigm and the analysis are sound, the results are convincing, and the interpretation is adequate. A particular strength of the study is the discovery-replication approach with replications of the findings in two independent datasets.

---

## [Referee Report · Reviewer #3 (Public review)]

Summary:

The authors suggest a new biomarker of chronic back pain with an option to predict a result of treatment.

Strengths:

The results were reproduced in three studies.

Weaknesses:

The number of participants is still low, an explanation of microstructure changes was not given, and some technical drawbacks are presented.

---

## [Author Response]

The following is the authors’ response to the original reviews.

**Public Reviews:**

**Reviewer #1 (Public Review):**
Summary:In this paper, Misic et al showed that white matter properties can be used to classify subacute back pain patients that will develop persisting pain.Strengths:Compared to most previous papers studying associations between white matter properties and chronic pain, the strength of the method is to perform a prediction in unseen data. Another strength of the paper is the use of three different cohorts. This is an interesting paper that provides a valuable contribution to the field.

We thank the reviewer for emphasizing the strength of our paper and the importance of validation on multiple unseen cohorts.

Weaknesses:The authors imply that their biomarker could outperform traditional questionnaires to predict pain: "While these models are of great value showing that few of these variables (e.g. work factors) might have significant prognostic power on the long-term outcome of back pain and provide easy-to-use brief questionnaires-based tools, (21, 25) parameters often explain no more than 30% of the variance (28-30) and their prognostic accuracy is limited.(31)". I don't think this is correct; questionnaire-based tools can achieve far greater prediction than their model in about half a million individuals from the UK Biobank (Tanguay-Sabourin et al., A prognostic risk score for the development and spread of chronic pain, Nature Medicine 2023).

We agree with the reviewer that we might have under-estimated the prognostic accuracy of questionnaire-based tools, especially, the strong predictive accuracy shown by Tangay-Sabourin 2023. In this revised version, we have changed both the introduction and the discussion to reflect the questionnaire-based prognostic accuracy reported in the seminal work by Tangay-Sabourin.

In the introduction (page 4, lines 3-18), we now write:

“Some studies have addressed this question with prognostic models incorporating demographic, pain-related, and psychosocial predictors.1-4 While these models are of great value showing that few of these variables (e.g. work factors) might have significant prognostic power on the long-term outcome of back pain, their prognostic accuracy is limited,5 with parameters often explaining no more than 30% of the variance.6-8. A recent notable study in this regard developed a model based on easy-to-use brief questionnaires to predict the development and spread of chronic pain in a variety of pain conditions capitalizing on a large dataset obtained from the UK-BioBank. 9 This work demonstrated that only few features related to assessment of sleep, neuroticism, mood, stress, and body mass index were enough to predict persistence and spread of pain with an area under the curve of 0.53-0.73. Yet, this study is unique in showing such a predictive value of questionnaire-based tools. Neurobiological measures could therefore complement existing prognostic models based on psychosocial variables to improve overall accuracy and discriminative power. More importantly, neurobiological factors such as brain parameters can provide a mechanistic understanding of chronicity and its central processing.”

And in the conclusion (page 22, lines 5-9), we write:

“Integrating findings from studies that used questionnaire-based tools and showed remarkable predictive power9 with neurobiological measures that can offer mechanistic insights into chronic pain development, could enhance predictive power in CBP prognostic modeling.”

Moreover, the main weakness of this study is the sample size. It remains small despite having 3 cohorts. This is problematic because results are often overfitted in such a small sample size brain imaging study, especially when all the data are available to the authors at the time of training the model (Poldrack et al., Scanning the horizon: towards transparent and reproducible neuroimaging research, Nature Reviews in Neuroscience 2017). Thus, having access to all the data, the authors have a high degree of flexibility in data analysis, as they can retrain their model any number of times until it generalizes across all three cohorts. In this case, the testing set could easily become part of the training making it difficult to assess the real performance, especially for small sample size studies.

The reviewer raises a very important point of limited sample size and of the methodology intrinsic of model development and testing. We acknowledge the small sample size in the “Limitations” section of the discussion. In the resubmission, we acknowledge the degree of flexibility that is afforded by having access to all the data at once. However, we also note that our SLF-FA based model is a simple cut-off approach that does not include any learning or hidden layers and that the data obtained from Open Pain were never part of the “training” set at any point at either the New Haven or the Mannheim site. Regarding our SVC approach we follow standard procedures for machine learning where we never mix the training and testing sets. The models are trained on the training data with parameters selected based on cross-validation within the training data. Therefore, no models have ever seen the test data set. The model performances we reported reflect the prognostic accuracy of our model. We write in the limitation section of the discussion (page 20, lines 20-21, and page 21, lines 1-6):

“In addition, at the time of analysis, we had “access” to all the data, which may lead to bias in model training and development. We believe that the data presented here are nevertheless robust since multisite validated but need replication. Additionally, we followed standard procedures for machine learning where we never mix the training and testing sets. The models were trained on the training data with parameters selected based on cross-validation within the training data. Therefore, no models have ever seen the test data set. The model performances we reported reflect the prognostic accuracy of our model”.

Finally, as discussed by Spisak et al., 10 the key determinant of the required sample size in predictive modeling is the ” true effect size of the brain-phenotype relationship”, which we think is the determinant of the replication we observe in this study. As such the effect size in the New Haven and Mannheim data is Cohen’s d >1.

Even if the performance was properly assessed, their models show AUCs between 0.65-0.70, which is usually considered as poor, and most likely without potential clinical use. Despite this, their conclusion was: "This biomarker is easy to obtain (~10 min of scanning time) and opens the door for translation into clinical practice." One may ask who is really willing to use an MRI signature with a relatively poor performance that can be outperformed by self-report questionnaires?

The reviewer is correct, the model performance is fair which limits its usefulness for clinical translation. We wanted to emphasize that obtaining diffusion images can be done in a short period of time and, hence, as such models’ predictive accuracy improves, clinical translation becomes closer to reality. In addition, our findings are based on older diffusion data and limited sample sizes coming from different sites and different acquisition sequences. This by itself would limit the accuracy especially since the evidence shows that sample size affects also model performance (i.e. testing AUC)10. In the revision, we re-worded the sentence mentioned by the reviewer to reflect the points discussed here. This also motivates us to collect a more homogeneous and larger sample. In the limitations section of the discussion, we now write (page 21, lines 6-9):

“Even though our model performance is fair, which currently limits its usefulness for clinical translation, we believe that future models would further improve accuracy by using larger homogenous sample sizes and uniform acquisition sequences.”

Overall, these criticisms are more about the wording sometimes used and the inference they made. I think the strength of the evidence is incomplete to support the main claims of the paper.Despite these limitations, I still think this is a very relevant contribution to the field. Showing predictive performance through cross-validation and testing in multiple cohorts is not an easy task and this is a strong effort by the team. I strongly believe this approach is the right one and I believe the authors did a good job.

We thank the reviewer for acknowledging that our effort and approach were useful.

Minor points:Methods:I get the voxel-wise analysis, but I don't understand the methods for the structural connectivity analysis between the 88 ROIs. Have the authors run tractography or have they used a predetermined streamlined form of 'population-based connectome'? They report that models of AUC above 0.75 were considered and tested in the Chicago dataset, but we have no information about what the model actually learned (although this can be tricky for decision tree algorithms).

We apologize for the lack of clarity; we did run tractography and we did not use a pre-determined streamlined form of the connectome.

Finding which connections are important for the classification of SBPr and SBPp is difficult because of our choices during data preprocessing and SVC model development: (1) preprocessing steps which included TNPCA for dimensionality reduction, and regressing out the confounders (i.e., age, sex, and head motion); (2) the harmonization for effects of sites; and (3) the Support Vector Classifier which is a hard classification model11.

In the methods section (page 30, lines 21-23) we added: “Of note, such models cannot tell us the features that are important in classifying the groups. Hence, our model is considered a black-box predictive model like neural networks.”

Minor:What results are shown in Figure 7? It looks more descriptive than the actual results.

The reviewer is correct; Figure 7 and Supplementary Figure 4 were both qualitatively illustrating the shape of the SLF. We have now changed both figures in response to this point and a point raised by reviewer 3. We now show a 3D depiction of different sub-components of the right SLF (Figure 7) and left SLF (Now Supplementary Figure 11 instead of Supplementary Figure 4) with a quantitative estimation of the FA content of the tracts, and the number of tracts per component. The results reinforce the TBSS analysis in showing asymmetry in the differences between left and right SLF between the groups (i.e. SBPp and SBPr) in both FA values and number of tracts per bundle.

**Reviewer #2 (Public Review):**
The present study aims to investigate brain white matter predictors of back pain chronicity. To this end, a discovery cohort of 28 patients with subacute back pain (SBP) was studied using white matter diffusion imaging. The cohort was investigated at baseline and one-year follow-up when 16 patients had recovered (SBPr) and 12 had persistent back pain (SBPp). A comparison of baseline scans revealed that SBPr patients had higher fractional anisotropy values in the right superior longitudinal fasciculus (SLF) than SBPp patients and that FA values predicted changes in pain severity. Moreover, the FA values of SBPr patients were larger than those of healthy participants, suggesting a role of FA of the SLF in resilience to chronic pain. These findings were replicated in two other independent datasets. The authors conclude that the right SLF might be a robust predictive biomarker of CBP development with the potential for clinical translation.Developing predictive biomarkers for pain chronicity is an interesting, timely, and potentially clinically relevant topic. The paradigm and the analysis are sound, the results are convincing, and the interpretation is adequate. A particular strength of the study is the discovery-replication approach with replications of the findings in two independent datasets.

We thank reviewer 2 for pointing to the strength of our study.

The following revisions might help to improve the manuscript further.- Definition of recovery. In the New Haven and Chicago datasets, SBPr and SBPp patients are distinguished by reductions of >30% in pain intensity. In contrast, in the Mannheim dataset, both groups are distinguished by reductions of >20%. This should be harmonized. Moreover, as there is no established definition of recovery (reference 79 does not provide a clear criterion), it would be interesting to know whether the results hold for different definitions of recovery. Control analyses for different thresholds could strengthen the robustness of the findings.

The reviewer raises an important point regarding the definition of recovery. To address the reviewers’ concern we have added a supplementary figure (Fig. S6) showing the results in the Mannheim data set if a 30% reduction is used as a recovery criterion, and in the manuscript (page 11, lines 1,2) we write: “Supplementary Figure S6 shows the results in the Mannheim data set if a 30% reduction is used as a recovery criterion in this dataset (AUC = 0.53)”.

We would like to emphasize here several points that support the use of different recovery thresholds between New Haven and Mannheim. The New Haven primary pain ratings relied on visual analogue scale (VAS) while the Mannheim data relied on the German version of the West-Haven-Yale Multidimensional Pain Inventory. In addition, the Mannheim data were pre-registered with a definition of recovery at 20% and are part of a larger sub-acute to chronic pain study with prior publications from this cohort using the 20% cut-off12. Finally, a more recent consensus publication13 from IMMPACT indicates that a change of at least 30% is needed for a moderate improvement in pain on the 0-10 Numerical Rating Scale but that this percentage depends on baseline pain levels.

- Analysis of the Chicago dataset. The manuscript includes results on FA values and their association with pain severity for the New Haven and Mannheim datasets but not for the Chicago dataset. It would be straightforward to show figures like Figures 1 - 4 for the Chicago dataset, as well.

We welcome the reviewer’s suggestion; we added these analyses to the results section of the resubmitted manuscript (page 11, lines 13-16): “The correlation between FA values in the right SLF and pain severity in the Chicago data set showed marginal significance (p = 0.055) at visit 1 (Fig. S8A) and higher FA values were significantly associated with a greater reduction in pain at visit 2 (p = 0.035) (Fig. S8B).”

- Data sharing. The discovery-replication approach of the present study distinguishes the present from previous approaches. This approach enhances the belief in the robustness of the findings. This belief would be further enhanced by making the data openly available. It would be extremely valuable for the community if other researchers could reproduce and replicate the findings without restrictions. It is not clear why the fact that the studies are ongoing prevents the unrestricted sharing of the data used in the present study.

We greatly appreciate the reviewer's suggestion to share our data sets, as we strongly support the Open Science initiative. The Chicago data set is already publicly available. The New Haven data set will be shared on the Open Pain repository, and the Mannheim data set will be uploaded to heiDATA or heiARCHIVE at Heidelberg University in the near future. We cannot share the data immediately because this project is part of the Heidelberg pain consortium, “SFB 1158: From nociception to chronic pain: Structure-function properties of neural pathways and their reorganization.” Within this consortium, all data must be shared following a harmonized structure across projects, and no study will be published openly until all projects have completed initial analysis and quality control.

**Reviewer #3 (Public Review):**
Summary:Authors suggest a new biomarker of chronic back pain with the option to predict the result of treatment. The authors found a significant difference in a fractional anisotropy measure in superior longitudinal fasciculus for recovered patients with chronic back pain.Strengths:The results were reproduced in three different groups at different studies/sites.Weaknesses:- The number of participants is still low.

The reviewer raises a very important point of limited sample size. As discussed in our replies to reviewer number 1:

We acknowledge the small sample size in the “Limitations” section of the discussion. In the resubmission, we acknowledge the degree of flexibility that is afforded by having access to all the data at once. However, we also note that our SLF-FA based model is a simple cut-off approach that does not include any learning or hidden layers and that the data obtained from Open Pain were never part of the “training” set at any point at either the New Haven or the Mannheim site. Regarding our SVC approach we follow standard procedures for machine learning where we never mix the training and testing sets. The models are trained on the training data with parameters selected based on cross-validation within the training data. Therefore, no models have ever seen the test data set. The model performances we reported reflect the prognostic accuracy of our model. We write in the limitation section of the discussion (page 20, lines 20-21, and page 21, lines 1-6):

“In addition, at the time of analysis, we had “access” to all the data, which may lead to bias in model training and development. We believe that the data presented here are nevertheless robust since multisite validated but need replication. Additionally, we followed standard procedures for machine learning where we never mix the training and testing sets. The models were trained on the training data with parameters selected based on cross-validation within the training data. Therefore, no models have ever seen the test data set. The model performances we reported reflect the prognostic accuracy of our model”.

Finally, as discussed by Spisak et al., 10 the key determinant of the required sample size in predictive modeling is the ” true effect size of the brain-phenotype relationship”, which we think is the determinant of the replication we observe in this study. As such the effect size in the New Haven and Mannheim data is Cohen’s d >1.

- An explanation of microstructure changes was not given.

The reviewer points to an important gap in our discussion. While we cannot do a direct study of actual tissue microstructure, we explored further the changes observed in the SLF by calculating diffusivity measures. We have now performed the analysis of mean, axial, and radial diffusivity.

In the results section we added (page 7, lines 12-19): “We also examined mean diffusivity (MD), axial diffusivity (AD), and radial diffusivity (RD) extracted from the right SLF shown in Fig.1 to further understand which diffusion component is different between the groups. The right SLF MD is significantly increased (p < 0.05) in the SBPr compared to SBPp patients (Fig. S3), while the right SLF RD is significantly decreased (p < 0.05) in the SBPr compared to SBPp patients in the New Haven data (Fig. S4). Axial diffusivity extracted from the RSLF mask did not show significant difference between SBPr and SBPp (p = 0.28) (Fig. S5).”

In the discussion, we write (page 15, lines 10-20):

“Within the significant cluster in the discovery data set, MD was significantly increased, while RD in the right SLF was significantly decreased in SBPr compared to SBPp patients. Higher RD values, indicative of demyelination, were previously observed in chronic musculoskeletal patients across several bundles, including the superior longitudinal fasciculus14. Similarly, Mansour et al. found higher RD in SBPp compared to SBPr in the predictive FA cluster. While they noted decreased AD and increased MD in SBPp, suggestive of both demyelination and altered axonal tracts,15 our results show increased MD and RD in SBPr with no AD differences between SBPp and SBPr, pointing to white matter changes primarily due to myelin disruption rather than axonal loss, or more complex processes. Further studies on tissue microstructure in chronic pain development are needed to elucidate these processes.”

- Some technical drawbacks are presented.

We are uncertain if the reviewer is suggesting that we have acknowledged certain technical drawbacks and expects further elaboration on our part. We kindly request that the reviewer specify what particular issues need to be addressed so that we can respond appropriately.

**Recommendations For The Authors:**

We thank the reviewers for their constructive feedback, which has significantly improved our manuscript. We have done our best to answer the criticisms that they raised point-by-point.

**Reviewer #2 (Recommendations For The Authors):**
The discovery-replication approach of the current study justifies the use of the terminus 'robust.' In contrast, previous studies on predictive biomarkers using functional and structural brain imaging did not pursue similar approaches and have not been replicated. Still, the respective biomarkers are repeatedly referred to as 'robust.' Throughout the manuscript, it would, therefore, be more appropriate to remove the label 'robust' from those studies.

We thank the reviewer for this valuable suggestion. We removed the label 'robust' throughout the manuscript when referring to the previous studies which didn’t follow the same approach and have not yet been replicated.

**Reviewer #3 (Recommendations For The Authors):**
This is, indeed, quite a well-written manuscript with very interesting findings and patient group. There are a few comments that enfeeble the findings.(1) It is a bit frustrating to read at the beginning how important chronic back pain is and the number of patients in the used studies. At least the number of healthy subjects could be higher.

The reviewer raises an important point regarding the number of pain-free healthy controls (HC) in our samples. We first note that our primary statistical analysis focused on comparing recovered and persistent patients at baseline and validating these findings across sites without directly comparing them to HCs. Nevertheless, the data from New Haven included 28 HCs at baseline, and the data from Mannheim included 24 HCs. Although these sample sizes are not large, they have enabled us to clearly establish that the recovered SBPr patients generally have larger FA values in the right superior longitudinal fasciculus compared to the HCs, a finding consistent across sites (see Figs. 1 and 3). This suggests that the general pain-free population includes individuals with both low and high-risk potential for chronic pain. It also offers one explanation for the reported lack of differences or inconsistent differences between chronic low-back pain patients and HCs in the literature, as these differences likely depend on the (unknown) proportion of high- and low-risk individuals in the control groups. Therefore, if the high-risk group is more represented by chance in the HC group, comparisons between HCs and chronic pain patients are unlikely to yield statistically significant results. Thus, while we agree with the reviewer that the sample sizes of our HCs are limited, this limitation does not undermine the validity of our findings.

(2) Pain reaction in the brain is in general a quite popular topic and could be connected to the findings or mentioned in the introduction.

We thank the reviewer for this suggestion. We have now added a summary of brain response to pain in general; In the introduction, we now write (page 4, lines 19-22 and page 5, lines 1-5):

“Neuroimaging research on chronic pain has uncovered a shift in brain responses to pain when acute and chronic pain are compared. The thalamus, primary somatosensory, motor areas, insula, and mid-cingulate cortex most often respond to acute pain and can predict the perception of acute pain16-19. Conversely, limbic brain areas are more frequently engaged when patients report the intensity of their clinical pain20, 21. Consistent findings have demonstrated that increased prefrontal-limbic functional connectivity during episodes of heightened subacute ongoing back pain or during a reward learning task is a significant predictor of CBP.12, 22. Furthermore, low somatosensory cortex excitability in the acute stage of low back pain was identified as a predictor of CBP chronicity.23”

(3) It is clearly observed structural asymmetry in the brain, why not elaborate this finding further? Would SLF be a hub in connectivity analysis? Would FA changes have along tract features? etc etc etc

The reviewer raises an important point. There is ground to suggest from our data that there is an asymmetry to the role of the SLF in resilience to chronic pain. We discuss this at length in the Discussion section. We have, in addition, we elaborated more in our data analysis using our Population Based Structural Connectome pipeline on the New Haven dataset. Following that approach, we studied both the number of fiber tracts making different parts of the SLF on the right and left side. In addition, we have extracted FA values along fiber tracts and compared the average across groups. Our new analyses are presented in our modified Figures 7 and Fig S11. These results support the asymmetry hypothesis indeed. The SLF could be a hub of structural connectivity. Please note however, given the nature of our design of discovery and validation, the study of structural connectivity of the SLF is beyond the scope of this paper because tract-based connectivity is very sensitive to data collection parameters and is less accurate with single shell DWI acquisition. Therefore, we will pursue the study of connectivity of the SLF in the future with well-powered and more harmonized data.

(4) Only FA is mentioned; did the authors work with MD, RD, and AD metrics?

We thank the reviewer for this suggestion that helps in providing a clearer picture of the differences in the right SLF between SBPr and SBPp. We have now extracted MD, AD, and RD for the predictive mask we discovered in Figure 1 and plotted the values comparing SBPr to SBPp patients in Fig. S3, Fig. S4., and Fig. S5 across all sites using one comprehensive harmonized analysis. We have added in the discussion “Within the significant cluster in the discovery data set, MD was significantly increased, while RD in the right SLF was significantly decreased in SBPr compared to SBPp patients. Higher RD values, indicative of demyelination, were previously observed in chronic musculoskeletal patients across several bundles, including the superior longitudinal fasciculus14. Similarly, Mansour et al. found higher RD in SBPp compared to SBPr in the predictive FA cluster. While they noted decreased AD and increased MD in SBPp, suggestive of both demyelination and altered axonal tracts15, our results show increased MD and RD in SBPr with no AD differences between SBPp and SBPr, pointing to white matter changes primarily due to myelin disruption rather than axonal loss, or more complex processes. Further studies on tissue microstructure in chronic pain development are needed to elucidate these processes.”

(5) There are many speculations in the Discussion, however, some of them are not supported by the results.

We agree with the reviewer and thank them for pointing this out. We have now made several changes across the discussion related to the wording where speculations were not supported by the data. For example, instead of writing (page 16, lines 7-9): “Together the literature on the right SLF role in higher cognitive functions suggests, therefore, that resilience to chronic pain is a top-down phenomenon related to visuospatial and body awareness.”, We write: “Together the literature on the right SLF role in higher cognitive functions suggests, therefore, that resilience to chronic pain might be related to a top-down phenomenon involving visuospatial and body awareness.”

(6) A method section was written quite roughly. In order to obtain all the details for a potential replication one needs to jump over the text.

The reviewer is correct; our methodology may have lacked more detailed descriptions. Therefore, we have clarified our methodology more extensively. Under “Estimation of structural connectivity”; we now write (page 28, lines 20,21 and page 29, lines 1-19):

“Structural connectivity was estimated from the diffusion tensor data using a population-based structural connectome (PSC) detailed in a previous publication.24 PSC can utilize the geometric information of streamlines, including shape, size, and location for a better parcellation-based connectome analysis. It, therefore, preserves the geometric information, which is crucial for quantifying brain connectivity and understanding variation across subjects. We have previously shown that the PSC pipeline is robust and reproducible across large data sets.24 PSC output uses the Desikan-Killiany atlas (DKA) 25 of cortical and sub-cortical regions of interest (ROI). The DKA parcellation comprises 68 cortical surface regions (34 nodes per hemisphere) and 19 subcortical regions. The complete list of ROIs is provided in the supplementary materials’ Table S6. PSC leverages a reproducible probabilistic tractography algorithm 26 to create whole-brain tractography data, integrating anatomical details from high-resolution T1 images to minimize bias in the tractography. We utilized DKA 25 to define the ROIs corresponding to the nodes in the structural connectome. For each pair of ROIs, we extracted the streamlines connecting them by following these steps: (1) dilating each gray matter ROI to include a small portion of white matter regions, (2) segmenting streamlines connecting multiple ROIs to extract the correct and complete pathway, and (3) removing apparent outlier streamlines. Due to its widespread use in brain imaging studies27, 28, we examined the mean fractional anisotropy (FA) value along streamlines and the count of streamlines in this work. The output we used includes fiber count, fiber length, and fiber volume shared between the ROIs in addition to measures of fractional anisotropy and mean diffusivity.”

(7) Why not join all the data with harmonisation in order to reproduce the results (TBSS)

We have followed the reviewer’s suggestion; we used neuroCombat harmonization after pooling all the diffusion weighted data into one TBSS analysis. Our results remain the same after harmonization.

In the Supplementary Information we added a paragraph explaining the method for harmonization; we write (SI, page 3, lines 25-34):

“Harmonization of DTI data using neuroCombat. Because the 3 data sets originated from different sites using different MR data acquisition parameters and slightly different recruitment criteria, we applied neuroCombat 29 to correct for site effects and then repeated the TBSS analysis shown in Figure 1 and the validation analyses shown in Figures 5 and 6. First, the FA maps derived using the FDT toolbox were pooled into one TBSS analysis where registration to a standard template FA template (FMRIB58_FA_1mm.nii.gz part of FSL) was performed. Next, neuroCombat was applied to the FA maps as implemented in Python with batch (i.e., site) effect modeled with a vector containing 1 for New Haven, 2 for Chicago, and 3 for Mannheim originating maps, respectively. The harmonized maps were then skeletonized to allow for TBSS.”

And in the results section, we write (page 12, lines 2-21):

“Validation after harmonization

Because the DTI data sets originated from 3 sites with different MR acquisition parameters, we repeated our TBSS and validation analyses after correcting for variability arising from site differences using DTI data harmonization as implemented in neuroCombat. 29 The method of harmonization is described in detail in the Supplementary Methods. The whole brain unpaired t-test depicted in Figure 1 was repeated after neuroCombat and yielded very similar results (Fig. S9A) showing significantly increased FA in the SBPr compared to SBPp patients in the right superior longitudinal fasciculus (MNI-coordinates of peak voxel: x = 40; y = - 42; z = 18 mm; t(max) = 2.52; p < 0.05, corrected against 10,000 permutations). We again tested the accuracy of local diffusion properties (FA) of the right SLF extracted from the mask of voxels passing threshold in the New Haven data (Fig.S9A) in classifying the Mannheim and the Chicago patients, respectively, into persistent and recovered. FA values corrected for age, gender, and head displacement accurately classified SBPr and SBPp patients from the Mannheim data set with an AUC = 0.67 (p = 0.023, tested against 10,000 random permutations, Fig. S9B and S7D), and patients from the Chicago data set with an AUC = 0.69 (p = 0.0068) (Fig. S9C and S7E) at baseline, and an AUC = 0.67 (p = 0.0098) (Fig. S9D and S7F) patients at follow-up, confirming the predictive cluster from the right SLF across sites. The application of neuroCombat significantly changes the FA values as shown in Fig.S10 but does not change the results between groups.”

Minor comments(1) In the case of New Haven data, one used MB 4 and GRAPPA 2, these two factors accelerate the imaging 8 times and often lead to quite a poor quality.Any kind of QA?

We thank the reviewer for identifying this error. GRAPPA 2 was in fact used for our T1-MPRAGE image acquisition but not during the diffusion data acquisition. The diffusion data were acquired with a multi-band acceleration factor of 4. We have now corrected this mistake.

(2) Why not include MPRAGE data into the analysis, in particular, for predictions?

We thank the reviewer for the suggestion. The collaboration on this paper was set around diffusion data. In addition, MPRAGE data from New Haven related to prediction is already published (10.1073/pnas.1918682117) and MPRAGE data of the Mannheim data set is a part of the larger project and will be published elsewhere.

(3) In preprocessing, the authors wrote: "Eddy current corrects for image distortions due to susceptibility-induced distortions and eddy currents in the gradient coil"However, they did not mention that they acquired phase-opposite b0 data. It means eddy_openmp works likely only as an alignment tool, but not susceptibility corrector.

We kindly thank the reviewer for bringing this to our attention. We indeed did not collect b0 data in the phase-opposite direction, however, eddy_openmp can still be used to correct for eddy current distortions and perform motion correction, but the absence of phase-opposite b0 data may limit its ability to fully address susceptibility artifacts. This is now noted in the Supplementary Methods under Preprocessing section (SI, page 3, lines 16-18): “We do note, however, that as we did not acquire data in the phase-opposite direction, the susceptibility-induced distortions may not be fully corrected.”

(4) Version of FSL?

We thank the reviewer for addressing this point that we have now added under the Supplementary Methods (SI, page 3, lines 10-11): “Preprocessing of all data sets was performed employing the same procedures and the FMRIB diffusion toolbox (FDT) running on FSL version 6.0.”

(5) Some short sketches about the connectivity analysis could be useful, at least in SI.

We are grateful for this suggestion that improves our work. We added the sketches about the connectivity analysis, please see Figure 7 and Supplementary Figure 11.

(6) Machine learning: functions, language, version?

We thank the reviewer for pointing out these minor points that we now hope to have addressed in our resubmission in the Methods section by adding a detailed description of the structural connectivity analysis. We added: “The DKA parcellation comprises 68 cortical surface regions (34 nodes per hemisphere) and 19 subcortical regions. The complete list of ROIs is provided in the supplementary materials’ Table S7. PSC leverages a reproducible probabilistic tractography algorithm 26 to create whole-brain tractography data, integrating anatomical details from high-resolution T1 images to minimize bias in the tractography. We utilized DKA 25 to define the ROIs corresponding to the nodes in the structural connectome. For each pair of ROIs, we extracted the streamlines connecting them by following these steps: (1) dilating each gray matter ROI to include a small portion of white matter regions, (2) segmenting streamlines connecting multiple ROIs to extract the correct and complete pathway, and (3) removing apparent outlier streamlines. Due to its widespread use in brain imaging studies27, 28, we examined the mean fractional anisotropy (FA) value along streamlines and the count of streamlines in this work. The output we used includes fiber count, fiber length, and fiber volume shared between the ROIs in addition to measures of fractional anisotropy and mean diffusivity.”

The script is described and provided at: https://github.com/MISICMINA/DTI-Study-Resilience-to-CBP.git.

(7) Ethical approval?

The New Haven data is part of a study that was approved by the Yale University Institutional Review Board. This is mentioned under the description of the data “New Haven (Discovery) data set (page 23, lines 1,2). Likewise, the Mannheim data is part of a study approved by Ethics Committee of the Medical Faculty of Mannheim, Heidelberg University, and was conducted in accordance with the declaration of Helsinki in its most recent form. This is also mentioned under “Mannheim data set” (page 26, lines 2-5): “The study was approved by the Ethics Committee of the Medical Faculty of Mannheim, Heidelberg University, and was conducted in accordance with the declaration of Helsinki in its most recent form.”

(1) Traeger AC, Henschke N, Hubscher M, et al. Estimating the Risk of Chronic Pain: Development and Validation of a Prognostic Model (PICKUP) for Patients with Acute Low Back Pain. PLoS Med 2016;13:e1002019.

(2) Hill JC, Dunn KM, Lewis M, et al. A primary care back pain screening tool: identifying patient subgroups for initial treatment. Arthritis Rheum 2008;59:632-641.

(3) Hockings RL, McAuley JH, Maher CG. A systematic review of the predictive ability of the Orebro Musculoskeletal Pain Questionnaire. Spine (Phila Pa 1976) 2008;33:E494-500.

(4) Chou R, Shekelle P. Will this patient develop persistent disabling low back pain? JAMA 2010;303:1295-1302.

(5) Silva FG, Costa LO, Hancock MJ, Palomo GA, Costa LC, da Silva T. No prognostic model for people with recent-onset low back pain has yet been demonstrated to be suitable for use in clinical practice: a systematic review. J Physiother 2022;68:99-109.

(6) Kent PM, Keating JL. Can we predict poor recovery from recent-onset nonspecific low back pain? A systematic review. Man Ther 2008;13:12-28.

(7) Hruschak V, Cochran G. Psychosocial predictors in the transition from acute to chronic pain: a systematic review. Psychol Health Med 2018;23:1151-1167.

(8) Hartvigsen J, Hancock MJ, Kongsted A, et al. What low back pain is and why we need to pay attention. Lancet 2018;391:2356-2367.

(9) Tanguay-Sabourin C, Fillingim M, Guglietti GV, et al. A prognostic risk score for development and spread of chronic pain. Nat Med 2023;29:1821-1831.

(10) Spisak T, Bingel U, Wager TD. Multivariate BWAS can be replicable with moderate sample sizes. Nature 2023;615:E4-E7.

(11) Liu Y, Zhang HH, Wu Y. Hard or Soft Classification? Large-margin Unified Machines. J Am Stat Assoc 2011;106:166-177.

(12) Loffler M, Levine SM, Usai K, et al. Corticostriatal circuits in the transition to chronic back pain: The predictive role of reward learning. Cell Rep Med 2022;3:100677.

(13) Smith SM, Dworkin RH, Turk DC, et al. Interpretation of chronic pain clinical trial outcomes: IMMPACT recommended considerations. Pain 2020;161:2446-2461.

(14) Lieberman G, Shpaner M, Watts R, et al. White Matter Involvement in Chronic Musculoskeletal Pain. The Journal of Pain 2014;15:1110-1119.

(15) Mansour AR, Baliki MN, Huang L, et al. Brain white matter structural properties predict transition to chronic pain. Pain 2013;154:2160-2168.

(16) Wager TD, Atlas LY, Lindquist MA, Roy M, Woo CW, Kross E. An fMRI-based neurologic signature of physical pain. N Engl J Med 2013;368:1388-1397.

(17) Lee JJ, Kim HJ, Ceko M, et al. A neuroimaging biomarker for sustained experimental and clinical pain. Nat Med 2021;27:174-182.

(18) Becker S, Navratilova E, Nees F, Van Damme S. Emotional and Motivational Pain Processing: Current State of Knowledge and Perspectives in Translational Research. Pain Res Manag 2018;2018:5457870.

(19) Spisak T, Kincses B, Schlitt F, et al. Pain-free resting-state functional brain connectivity predicts individual pain sensitivity. Nat Commun 2020;11:187.

(20) Baliki MN, Apkarian AV. Nociception, Pain, Negative Moods, and Behavior Selection. Neuron 2015;87:474-491.

(21) Elman I, Borsook D. Common Brain Mechanisms of Chronic Pain and Addiction. Neuron 2016;89:11-36.

(22) Baliki MN, Petre B, Torbey S, et al. Corticostriatal functional connectivity predicts transition to chronic back pain. Nat Neurosci 2012;15:1117-1119.

(23) Jenkins LC, Chang WJ, Buscemi V, et al. Do sensorimotor cortex activity, an individual's capacity for neuroplasticity, and psychological features during an episode of acute low back pain predict outcome at 6 months: a protocol for an Australian, multisite prospective, longitudinal cohort study. BMJ Open 2019;9:e029027.

(24) Zhang Z, Descoteaux M, Zhang J, et al. Mapping population-based structural connectomes. Neuroimage 2018;172:130-145.

(25) Desikan RS, Segonne F, Fischl B, et al. An automated labeling system for subdividing the human cerebral cortex on MRI scans into gyral based regions of interest. Neuroimage 2006;31:968-980.

(26) Maier-Hein KH, Neher PF, Houde J-C, et al. The challenge of mapping the human connectome based on diffusion tractography. Nature Communications 2017;8:1349.

(27) Chiang MC, McMahon KL, de Zubicaray GI, et al. Genetics of white matter development: a DTI study of 705 twins and their siblings aged 12 to 29. Neuroimage 2011;54:2308-2317.

(28) Zhao B, Li T, Yang Y, et al. Common genetic variation influencing human white matter microstructure. Science 2021;372.

(29) Fortin JP, Parker D, Tunc B, et al. Harmonization of multi-site diffusion tensor imaging data. Neuroimage 2017;161:149-170.